# Conformational disorder of organic cations tunes the charge carrier mobility in two-dimensional organic-inorganic perovskites

Chuanzhao Li[1], Jin Yang[2,3], Fuhai Su[2], Junjun Tan[1], Yi Luo [1] & Shuji Ye [1✉]

The chemical nature of the organic cations governs the optoelectronic properties of two-dimensional organic-inorganic perovskites. But its mechanism is not fully understood. Here, we apply femtosecond broadband sum frequency generation vibrational spectroscopy to investigate the molecular conformation of spacer organic cations in two-dimensional organic-inorganic perovskite films and establish a correlation among the conformation of the organic cations, the charge carrier mobility, and broadband emission. Our study indicates that both the mobility and broadband emission show strong dependence on the molecular conformational order of organic cations. The gauche defect and local chain distortion of organic cations are the structural origin of the in-plane mobility reduction and broad emission in two-dimensional organic-inorganic perovskites. Both of the interlayer distance and the conformational order of the organic cations affect the out-of-plane mobility. This work provides molecular-level understanding of the conformation of organic cations in optimizing the optoelectronic properties of two-dimensional organic-inorganic perovskites.

[1] Hefei National Laboratory for Physical Sciences at the Microscale, Department of Chemical Physics, and Synergetic Innovation Center of Quantum Information & Quantum Physics, University of Science and Technology of China, 230026 Hefei, China. [2] Key Laboratory of Materials Physics, Institute of Solid State Physics, Hefei Institutes of Physical Science, Chinese Academy of Sciences, 230031 Hefei, China. [3] University of Science and Technology of China, 230026 Hefei, China. ✉email: shujiye@ustc.edu.cn

Two-dimensional (2D) organic–inorganic hybrid perovskites (OIHPs) have been demonstrated to possess excellent photophysical properties[1], wider structural diversity[2], and superior stability[3]. These characteristics make OIHPs be a promising candidate for nanophotonics and optoelectronic devices, such as solar cells[4], light-emitting diodes (LEDs)[5], field effect transistors (FETs)[6], photodetectors[7], and lasers[8]. 2D OIHPs (general formula $A_2MX_4$) can be viewed as natural multiple-quantum-wells in which 2D semiconducting layers ($[MX_4]^{2-}$) are separated by organic spacer cations ($A^+$)[9,10]. Many studies have shown that chemical tuning of spacer organic cations can provide a simple but effective approach to improve the stability and performance of 2D OIHPs[11–13]. For instance, fine selection of the spacer cations can achieve sharp photoluminescence (PL) spectra and the PL quantum yield up to 79% in the blue region[14]. Although important milestones have been achieved through this chemical engineering[9], how the chemical nature of the organic cations within the inorganic framework governs the structural and optoelectronic properties of 2D OIHPs is not fully understood. More specifically, while such performance is strongly determined by the charge-carrier mobility ($\mu$), the excited carrier lifetime ($\tau$), and emission characteristics[15,16], several views on how the structural parameters of spacer cations affect the charge-carrier mobility and PL emission have been proposed[11,17–19]. Some studies suggest that increasing the length of organic cations can cause structural reorganization, thus leading to the formation of 1D quantum confinement effect[17,20], while others claim that the charge transport property of 2D OIHP films can be efficiently improved by reducing the interlayer distance and further giving rise to better device performance[18]. Moreover, it is also implied that interlayer distance and barrier height/structure coequally control the charge transport[19]. In-depth insights into the structural origins of the carrier dynamics and emission characteristics hold the key for understanding the mechanism of remarkable performances of 2D OIHPs and thus driving the design and discovery of new functional materials.

In this study, we synthesize a series of 2D OIHP films with large organic spacer cations ($[CH_3(CH_2)_{n-1}NH_3]_2PbI_4$, $n = 4, 6, 8, 10, 12, 18$) using a common procedure[21] and systematically investigate the influence of the molecular structures of cations on their carrier mobility and PL properties. A remarkable dependence of carrier mobility and broad PL emission on the conformational order of the alkyl chains in organic cations is revealed. Such finding is achieved thanks to the application of sum frequency generation vibrational spectroscopy (SFG-VS), supplemented by optical-pump terahertz-probe spectroscopy (OPTPS), current-voltage (I–V) measurements, temperature-dependent PL spectroscopy, and X-ray diffraction (XRD) measurements. SFG-VS is a second-order coherent nonlinear optical technique that is quite sensitive to molecular symmetry, molecular order, and orientation[22–27]. Its selection rules dictate that signals solely arise from non-centrosymmetric motifs, making it a powerful and versatile tool to identify the molecular conformation within alkyl chains and track microstructural changes of the organic cations[28–30]. Our results indicate that the so far largely ignored molecular order within $[PbI_6]^{4-}$-bound organic cations plays a critical role in governing the electron–phonon interactions, thus leading to mobility changes and broadband emission. This study highlights the importance of molecular-level structural engineering in hybrid perovskites to control the optoelectronic properties.

## Results

### The charge-carrier mobility and broadband emission. 
We determine the in-plane charge-carrier mobility of 2D OIHP films using OPTPS experiments. OPTPS is a technique that allows to determine the charge-carrier mobility along the film plane. It is only sensitive to the presence of free charge carriers. Trapped charges or neutral Coulomb-bound electron–hole pairs (excitons) will not be detected[31]. This technique has been demonstrated to be a powerful tool for determining the effective charge-carrier mobility along the film plane ($\varphi\mu_i$)[12,32–37]. Here $\varphi$ is the photon-to-charge branching ratio. It has been applied to measure the carrier mobility of a series of perovskites (See Supplementary Table 1). According to the previous studies[12,32–37], the value of effective charge-carrier mobility ($\varphi\mu_i$) can be deduced from the change in THz amplitude ($\Delta T/T$) (see Supplementary Note 1). Figure 1a displays the transient THz transmission dynamics of $HA_2PbI_4$ ($n = 6$) following photoexcitation with the excited wavelength at 400 nm at normal incidence angle and with pump fluences ranging from 16 to 73.6 $\mu$J cm$^{-2}$. The data for other five 2D OIHP films are given in Supplementary Fig. 1. The decay kinetics (Supplementary Fig. 1) can be modeled by a biexponential function (Supplementary Note 2) and gives a fast component $\tau_1$ of 0.8–1.2 ps with the amplitude ratio about 50% and a slow component $\tau_2$ of ~10 ps, respectively. It is found that the fast component $\tau_1$ displays independence of pump intensity while the slow component $\tau_2$ decreases as the pump fluence increases (Supplementary Fig. 2 and Supplementary Note 3). These two components are attributed to the electron–phonon scattering process ($\tau_1$) and many-body interactions such as Auger process ($\tau_2$)[38], respectively. Previous studies have indicated that phonon-mediated relaxation time falls in few hundreds of femtoseconds in a few-layer 2D materials such as $MoS_2$[38].

Supplementary Fig. 3 shows the maximum of $|\Delta T/T|$ of the six 2D OIHP thin films as a function of excitation fluence. When the fluence is less than 45 $\mu$J cm$^{-2}$, there is a linear correlation between the maximum of $|\Delta T/T|$ and the excitation fluence. The maximum of $|\Delta T/T|$ does not increase linearly with pump fluence of >45 $\mu$J cm$^{-2}$, suggesting the presence of two-photon absorption or nesting effects. To avoid these unwelcome effects, we only choose the $|\Delta T/T|$ at a pump fluence of 25.6 $\mu$J cm$^{-2}$ to extract the mobility value of $\varphi\mu_i$ (Fig. 1b). The values of $\varphi\mu_i$ are determined to be 0.54, 1.49, 1.1, 0.89, 0.82, 0.45 cm$^2$ V$^{-1}$ s$^{-1}$ for $n = 4, 6, 8, 10, 12, 18$, respectively, which are in the same magnitude as previous report[32]. It is found that the values of $\varphi\mu_i$ initially increase ($n \leq 6$) and then decrease ($n \geq 6$) with the chain length of the organic cations increasing. $HA_2PbI_4$ ($n = 6$) has the largest effective charge-carrier mobility. It is worth mentioning that the difference in the $\varphi$ value may affect the charge-carrier mobility deduced by OPTPS experiments. In principle, the value of $\varphi$ is determined by the exciton binding energy ($E_b$) of the materials[39]. The value of $E_b$ of 2D OIHPs of $[CH_3(CH_2)_{n-1}NH_3]_2PbI_4$ ($n = 4, 6, 8, 9, 10,$ and $12$) has been measured by Ishihara et al. in terms of the difference between the bandgap and excitonic absorption peak in the low-temperature (5 K) absorption spectrum[40]. They found that the value is ~320 meV and is independent of the length of the alkyl chains of the organic cations. This conclusion has also been confirmed by theoretical calculation[41]. According to previous reports, $\varphi$ can be assumed to be similar for all the studied 2D OIHPs in this work.

We then determine out-of-plane charge-carrier mobility ($\mu_o$) using Mott–Gurney (M–G) analysis of the I–V data curves (Supplementary Fig. 4). M–G analysis of I–V data has been used to estimate the carrier mobility (Supplementary Note 4)[18,42]. The pictures of real devices based on six 2D OIHPs have been shown in Supplementary Fig. 5. As shown in Fig. 1c and Supplementary Table 2, out-of-plane charge-carrier mobility is about four orders of magnitude smaller than the in-plane mobility and it follows similar trend of in-plane mobility. Namely, out-of-plane charge-carrier mobility also initially increases ($n \leq 6$) and then decreases ($n \geq 6$) as the chain length of the organic cations increases.

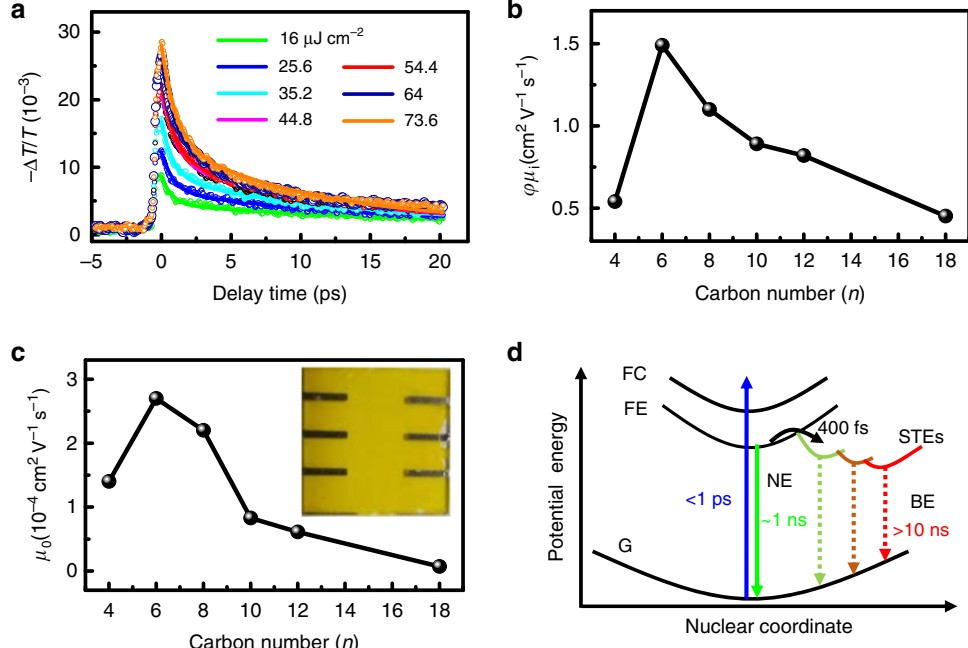

**Fig. 1 Charge-carrier mobility determined by OPTPS measurements and current-voltage (I-V) measurements. a** Representative transient THz transmission changes, ΔT/T, of $HA_2PbI_4$ ($n = 6$) for different excitation fluences ($\lambda_{pump} = 400$ nm). Symbols represent experimental data while solid lines are fitting to the data using biexponential function. **b** Effective charge-carrier mobility along the film plane ($\varphi\mu_i$) determined by OPTPS with the pump fluence of 25.6 μJ cm$^{-2}$. **c** Out-of-plane charge-carrier mobility ($\mu_o$) determined by Mott–Gurney analysis of the I-V data curves. Inset shows a picture of the devices/films. **d** Schematic of the adiabatic potential energy curves of the ground state (G), free-exciton state (FE), free-carrier state (FC), and various self-trapped excited states (STEs). The vertical dashed line shows possible nonradiative decay processes of the FE and STEs.

In theory, the charge-carrier mobility is governed by intrinsic and extrinsic effects[37,43–45]. The intrinsic effects originate from the charge-carrier interactions with the lattice (or phonons), while the extrinsic effects arise from the material imperfections, such as impurities/dopants, and grain boundaries. All of these effects could cause the charge-carrier mobility to decrease. In current study, the charge-carrier mobility is dominated by in-plane mobility derived from OPTPS which represents a lower bound for the actual mobility and is the summation over hole and electron mobilities[12,36,37]. In principle, the carrier transport probed by OPTPS mostly reflects intrinsic material properties, rather than extrinsic factors[12,32,37]. According to the Drude model[37,43,46], the strong electron–phonon scattering/coupling could reduce the carrier mobility due to polaron formation and charge-carrier trapping, which could dramatically impact the fluctuation of charge-carrier transport. On the other hand, the presence of self-trapped states can be manifested by the broadband emission below the bandgap[31,47]. As illustrated in Fig. 1d, after free carriers (FCs) are relaxed to free excitons (FEs), a manifold of self-trapped excited states (STEs) with different energies lying within the bandgap will gradually form[31,48]. Thus, the radiative decay from the STE potential well gives rise to broadband PL. Since the self-trapped carrier has been found to be largely dependent on the dimensionality of materials[49,50], it can be naturally anticipated that the mobility difference shown in Fig. 1b most likely contributes to the formation of self-trapped carriers in 2D OIHPs. There will thus be a correlation between the effective charge-carrier mobility and broadband emission. To verify this conjecture, we measure the low-temperature PL spectra for these six 2D OIHP polycrystalline thin films. Low-temperature PL measurement is used because the broadband emission at room temperature in 2D OIHPs is very weak due to unavoidable presence of scattering[50]. Recently, Wu et al. have confirmed that the existence of STE state can be determined by low-temperature

PL spectroscopy even though low-temperature phase transition may shift the PL peak position. They have demonstrated that the result deduced from low-temperature PL emission is consistent with the one measured by transient absorption spectroscopy at room temperature[50]. Figure 2a–c displays temperature-dependent PL spectral maps of $BA_2PbI_4$ ($n = 4$), $HA_2PbI_4$ ($n = 6$), and $OA_2PbI_4$ ($n = 8$). The spectral maps of $DA_2PbI_4$ ($n = 10$), $DDA_2PbI_4$ ($n = 12$), and $ODA_2PbI_4$ ($n = 18$) are shown in Supplementary Fig. 6. All of the samples exhibit narrow emission (NE) at the bandgap energy (480–540 nm). This NE corresponds to free-exciton emission upon photoexcitation. Besides the narrow emission, a broad emission (BE) is observed at the temperature ranging from 80 K to 150 K for the samples with the exception of $HA_2PbI_4$ ($n = 6$) (Supplementary Fig. 7). Such spectral features are consistent with previous reports, in which the film of $n = 6$ does not have an obvious broadband PL peak at above 100 K, while the films of $n = 4$ and $n = 12$ have strong broadband PL signals at 100 K[50–52].

As mentioned above, the self-trapped states are associated with the broad emission[31,47]. The ratio of integrated intensity between BE and NE ($I_{BE}/I_{NE}$) is indicative of an equilibrium between "free" and "self-trapped" emissive states[31,47]. To qualitatively analyze this ratio, we fit the PL spectra using Gaussian functions (Supplementary Fig. 8 and Supplementary Note 5). The temperature-dependent ratio at T = 80–150 K is shown in Fig. 2d. The decrease in the $I_{BE}/I_{NE}$ ratio with increasing temperature is because the self-trapped states can be thermally activated into the free-exciton states[47,50]. The broadband PL emission is very weak at room temperature, indicating the radiative recombination is small and the nonradiative recombination is dominant at room temperature. Broadband emission becomes apparent at low temperature, illustrating the attenuation of nonradiative decay channels and enhancement of the radiative recombination[31]. Figure 2e plots the $I_{BE}/I_{NE}$ ratio at 80 K against the chain length of

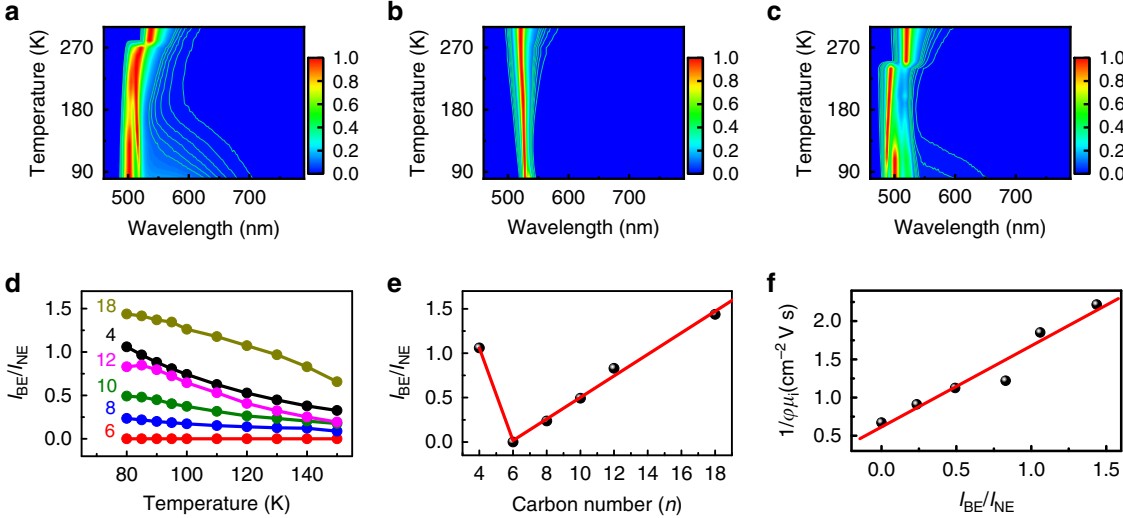

**Fig. 2 Temperature-dependent fluorescence map of 2D OIHP films obtained by heating from 80 K to room temperature. a** BA$_2$PbI$_4$ ($n = 4$), **b** HA$_2$PbI$_4$ ($n = 6$), and **c** OA$_2$PbI$_4$ ($n = 8$). **d** Ratio of integrated BE to NE PL intensity ($I_{BE}/I_{NE}$) of six samples as a function of temperature range from 80 K to 150 K. **e** The $I_{BE}/I_{NE}$ ratio at 80 K is plotted against the alkyl chain length of organic cations. **f** The correlation between the $I_{BE}/I_{NE}$ ratio at 80 K and the reciprocal of the in-plane mobility.

the organic cations. It is found that the ratio initially decreases ($n \leq 6$) and then increases ($n \geq 6$) with the chain length of the organic cations increasing. This is exactly what we have predicted, i.e., a linear correlation between the $I_{BE}/I_{NE}$ ratio and the reciprocal of the in-plane mobility is clearly observed (Fig. 2f), revealing that self-trapped states could deteriorate the in-plane mobility of 2D OIHP thin films as the strong electron–phonon coupling occurs.

**Molecular conformation of spacer organic cations**. It is noticed that these six samples have identical chemical composition of inorganic cage (i.e., the Pb–I layer), the difference in the effective charge-carrier mobility and broadband emission should originate from the packing of the organic cations, which may cause distortion of the inorganic cage. We then examine molecular conformation of the 2D OIHP thin films at room temperature using femtosecond broadband SFG-VS, to look for correlations between the mobility, broad emission and the structures of organic cations. Owing to the unique structure of 2D OIHPs, involving polar organic molecules with many degrees of freedom embedded within a rather rigid inorganic Pb–I framework, motions of the organic component can greatly affect the perovskite photovoltaic properties such as conductivity and mobility[53]. For instance, rotation and other rearrangement of the highly anisotropic and polar organic molecules can "solvate" the charge carriers, and thus screen the interaction of charge carriers with longitudinal optical (LO) phonons, which is accompanied by deformation of the inorganic framework[54,55]. Unfortunately, molecular structures of [PbI$_6$]$^{4-}$-bound organic molecules are not well understood. Due to lack of crystallinity, very few techniques can directly provide insights into the intermolecular structures of ligands on the inorganic framework [PbI$_6$]$^{4-}$ within films. Here, we present the application of symmetry-sensitive SFG as a direct probe for molecular conformation of the organic cations. Because of the ordered arrangement of the lattice of 2D OIHPs, the SFG signals of OIHPs are actually similar to the case of quartz, in which SFG signals come from not only the molecules at surface and interface, but also the bulk molecules. Such selection rules make SFG show powerful potential in the research of nanomaterials. In fact, SFG has been applied to probe molecular conformation of the organic

cations of lead-halide perovskite[23,56] and the capping ligands on nanoparticle surfaces[24,25,30].

In terms of SFG selection rules, in well-ordered and all-trans-configured alkyl chains, the SFG signals are dominated by the terminal CH$_3$ groups, little or no contribution from CH$_2$ stretching modes because CH$_2$ groups adopt a near inversion symmetry configuration. In contrast, gauche defects along the carbon backbone of organic cations can disrupt the local inversion centers, which makes the CH$_2$ stretching modes become SFG active. Therefore, the amplitude ratio of the symmetric methyl (ss-CH$_3$) and methylene (ss-CH$_2$) stretches ($\chi^{(2)}_{ss-CH_3}/\chi^{(2)}_{ss-CH_2}$) has been proven to be an effective optical ruler for evaluating the gauche defect and local chain distortion of n-alkyl chains[24–26]. In general, lower ratio indicates that the alkyl chains are more disordered. Figure 3a shows the ssp SFG spectra of all six 2D OIHP films with an average thickness of 144 nm on SiO$_2$ prisms in the CH-stretch region ranging from 2800 to 3025 cm$^{-1}$. The ppp spectra are put in Supplementary Fig. 9a. The spectra are dominated by the signals from the CH$_2$ symmetric stretch (ss-CH$_2$, ~2850 cm$^{-1}$), CH$_3$ symmetric stretch (ss-CH$_3$, ~2875 cm$^{-1}$), CH$_2$ asymmetric stretch (as-CH$_2$, ~2912 cm$^{-1}$), Fermi resonance of CH$_3$ group (Fermi-CH$_3$, ~2940 cm$^{-1}$), and CH$_3$ asymmetric stretch (as-CH$_3$, ~2965 cm$^{-1}$)[24–26]. To quantitatively deduce the $\chi^{(2)}_{ss-CH_3}/\chi^{(2)}_{ss-CH_2}$ ratio, we fit the SFG spectra using a standard procedure (Supplementary Note 6). The $\chi^{(2)}_{ss-CH_3}/\chi^{(2)}_{ss-CH_2}$ ratio is found to show the same trend with the effective charge-carrier mobility, namely, it initially increases ($n \leq 6$) and then decreases ($n \geq 6$) with the chain length of the organic cations increasing (Fig. 3b). HA$_2$PbI$_4$ ($n = 6$) has the least gauche defect of alkyl chain. It is evident that as the $\chi^{(2)}_{ss-CH_3}/\chi^{(2)}_{ss-CH_2}$ ratio increases, the effective in-plane mobility ($\varphi\mu_i$) linearly increases while the $I_{BE}/I_{NE}$ ratio linearly decreases (Fig. 3c). This correlation implies that the gauche defect and local chain distortion of organic cations are the structural origin of the in-plane mobility reduction and broad emission in these six 2D OIHP films. The presence of gauche defect and local chain distortion can form a potential well that can underpin the localization and self-trapping of charge carriers, leading to lower charge transport along the plane and greater broadband PL due to stronger electron–phonon coupling. We therefore conclude that the generation of gauche defect and local

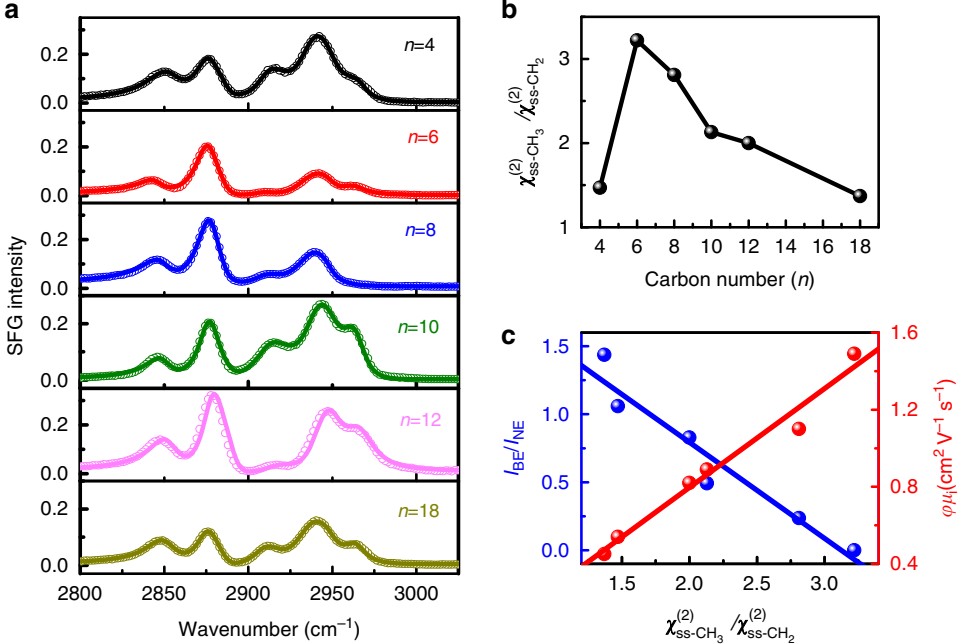

**Fig. 3 The structure-property relationship in the 2D OIHP thin films with a thickness of 144 nm. a** ssp-polarized SFG spectra of six samples, where symbols represent experimental data while solid lines are fitted data, respectively. **b** The $\chi^{(2)}_{ss-CH_3}/\chi^{(2)}_{ss-CH_2}$ ratio is plotted against the alkyl chain length of organic cations. **c** The $I_{BE}/I_{NE}$ ratio at 80 K and the in-plane mobility correlate with the $\chi^{(2)}_{ss-CH_3}/\chi^{(2)}_{ss-CH_2}$ ratio.

chain distortion can reduce the in-plane mobility. Indeed, the mobility reduced by gauche defect and local chain distortion has been theoretically predicted[57] and experimentally observed in organic thin film transistor[26].

There are some concerns that the $\chi^{(2)}_{ss-CH_3}/\chi^{(2)}_{ss-CH_2}$ ratio may be affected by the chain orientation and the film thickness. Theoretically, the ratio of $\chi^{(2)}_{ss-CH_3}/\chi^{(2)}_{ss-CH_2}$ not only depends on the fraction of defected chains but also relies on the orientation of the hydrocarbon chain[28,29]. However, previous studies have demonstrated that a dramatic reduction in the order ratio without introduction of more defects must require a significant variation of the orientation angle (see Supplementary Note 7)[30,58,59], which is not the case of our observations. In our study, it is found that the average orientation of the terminal methyl groups changes very small for these six 2D OIHP films because the measured ppp and ssp spectral intensity ratio $(\chi^{(2)}_{ppp}(CH_3, ss)/\chi^{(2)}_{ssp}(CH_3, ss))$ is almost the same $(0.9 \pm 0.1)$ (Supplementary Figs. 9b, 10d, and 11d). Herein, the ratio of $\chi^{(2)}_{ss-CH_3}/\chi^{(2)}_{ss-CH_2}$ is dominated by the defected chains and not the chain orientation. To evaluate the influence of the film thickness, we further measure the SFG spectra of the films with the thickness of 9.0 nm and 55 nm (Supplementary Figs. 10 and 11). It is found that although the $\chi^{(2)}_{ss-CH_3}/\chi^{(2)}_{ss-CH_2}$ ratio has a little variation for different thicknesses, the dependence of the $\chi^{(2)}_{ss-CH_3}/\chi^{(2)}_{ss-CH_2}$ ratio on the alkyl chain length follows similar trend for the films with the same thickness. It is noted that different vibrational modes of organic or polymer films in window geometry have been reported to be affected by many factors including substrate-molecule interaction[60], phase transition[61], and film thickness[62,63]. However, in our experiments, we employ a near-total-internal-reflection geometry (SiO$_2$ prisms) to collect SFG spectra at room temperature. Under this experimental geometry, symmetric CH$_2$ and CH$_3$ modes have almost the same Fresnel coefficient. In addition, the SFG signals of OIHPs actually come from total

asymmetry of the films (including the bulk molecules) because of the ordered arrangement of the lattice of 2D OIHPs. Furthermore, phase transition does not occur to these samples at room temperature. Therefore, the ratio obtained in our sample system shows very small dependence on the film thickness. Finally, the surface roughness determined by AFM is $2.0 \pm 1.0$ nm for the films of $n = 4$–8 and $10.0 \pm 2.0$ nm for the films of $n = 10$–18. There is no correlation between the alkyl chain conformation and the roughness (Supplementary Fig. 12).

It needs to mention that some studies claimed that the charge-carrier mobility of 2D OIHPs increases as the interlayer distance decreases[18]. To examine the relationship between mobility and interlayer distance, we determine the interlayer distance of these 2D OIHP thin films by using XRD techniques in terms of the positions of (002) diffraction peak (Fig. 4a). The peak positions agree well with the values modeled using the software of VESTA[64–66] (Supplementary Fig. 13 and Supplementary Table 3). It can be seen that the (002) diffraction peaks display a shift toward smaller angles as the organic length increases (from BA$_2$PbI$_4$ ($n = 4$) to ODA$_2$PbI$_4$ ($n = 18$)). Based on Bragg's law, the distance (d) between the inorganic layers for [C$_n$H$_{2n+1}$NH$_3$]$_2$PbI$_4$ films is determined to be 1.36 nm ($n = 4$), 1.66 nm ($n = 6$), 1.83 nm ($n = 8$), 2.08 nm ($n = 10$), 2.38 nm ($n = 12$), and 3.08 nm ($n = 18$), respectively. The interlayer distance linearly increases with the chain length of the organic cations increasing (Fig. 4b). Figure 5a, b plots the charge-carrier mobility against the interlayer distance. It is evident that both of in-plane and out-of-plane mobilities decrease as the interlayer distance increases for the samples from HA$_2$PbI$_4$ ($n = 6$) to ODA$_2$PbI$_4$ ($n = 18$), which is in good agreement with previous studies[18,67,68]. However, the sample of BA$_2$PbI$_4$ ($n = 4$) does not satisfy such law. In contrast, out-of-plane mobility shows a linear correlation with the ratio of $(\chi^{(2)}_{ss-CH_3}/\chi^{(2)}_{ss-CH_2})/d$ (Fig. 5c). It implies that out-of-plane mobility is not only related to the interlayer distance but also dependent on the conformational order of the organic cations. The relationships in Fig. 3c and Fig. 5c reveal that the

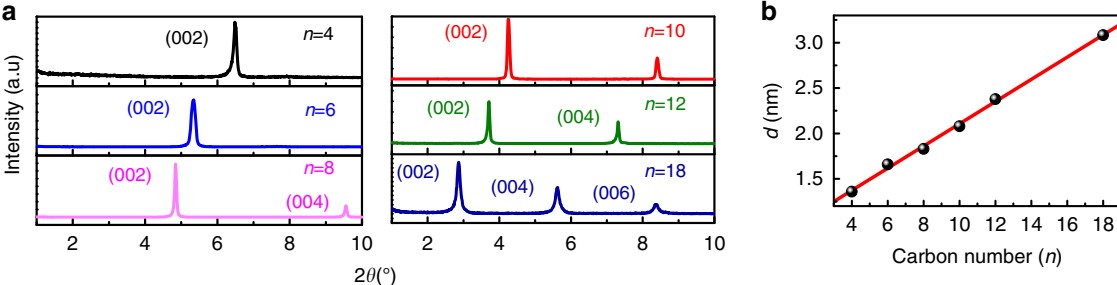

**Fig. 4 The XRD results of six 2D OIHP thin films. a** XRD patterns. **b** The dependence of interlayer distance (d) on the alkyl chain length of organic cations. The interlayer distance is calculated in terms of the positions of (002) diffraction peak.

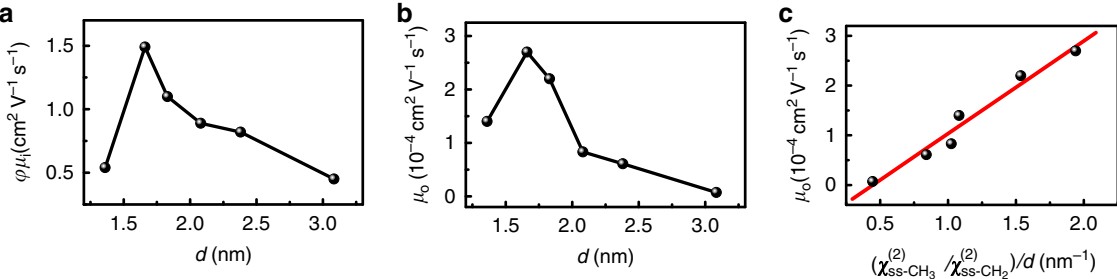

**Fig. 5 The charge-carrier mobility is plotted against the interlayer distance or the ratio of $(\chi^{(2)}_{ss-CH_3}/\chi^{(2)}_{ss-CH_2})/d$. a** The effective in-plane mobility versus interlayer distance. **b** The out-of-plane mobility versus interlayer distance. **c** The out-of-plane mobility versus the ratio of $(\chi^{(2)}_{ss-CH_3}/\chi^{(2)}_{ss-CH_2})/d$.

conformational order of the organic cations plays important roles in regulating the charge-carrier mobility and broadband emission in 2D OIHP films.

## Discussion

We have applied femtosecond broadband SFG-VS to investigate the molecular conformation of spacer organic cations in 2D OIHP films and established a correlation among the conformation of the organic cations, the charge-carrier mobility, and broadband emission. It is found that both the mobility and broadband emission show strong dependence on the molecular conformational order of organic cations. It implies that the gauche defect and local chain distortion of organic cations are the structural origin of the in-plane mobility reduction and broad emission in 2D OIHP films. The interlayer distance and the conformational order of the organic cations coequally regulate the out-of-plane mobility. Of course, the formation of gauche defects in the alkyl chain may cause a slight local distortion of the perovskite lattice. Theoretical simulations are highly desirable to confirm it in the future. Femtosecond visible pump-IR probe will also be helpful to reveal the coupling between inorganic anions and organic cations. Anyway, this work provides physical understanding of the important role of organic cation conformation in optimizing the optoelectronic properties of 2D OIHPs. In addition, our work also highlights the power of the state-of-the-art SFG-VS technique in revealing structure-property relationship in the perovskite research at the molecular level.

## Methods

**Materials.** N, N-dimethylformamide (DMF, purity ≥99.5%), N, N-dimethylformamide-d$_7$ (DMF-d$_7$, purity ≥99.5%) were purchased from Sigma-Aldrich. Lead iodide (PbI$_2$, purity ~99%). Butylammonium iodide (BAI, purity ~99%), hexylammonium iodide (HAI, purity ~99%), octylammonium iodide (OAI, purity ~99%), decylammonium iodide (DAI, purity ~99%), dodecylammonium iodide (DDAI, purity ~99%), and octadecylammonium iodide (ODAI, purity ~99%) were all purchased from Xi'An Polymer Light Technology Corp (Xi'An, China). All the chemicals were used without further purification.

**Synthesis of 2D OIHP thin films.** The 2D OIHPs were synthesized using a common procedure in terms of previous reports[21]. In brief, $C_nH_{2n+1}NH_3I$ and PbI$_2$ (with the molar ratio of 2:1) were first dissolved in DMF (or DMF-d$_7$) to prepare the perovskite precursor solutions. The 2D OIHP films were prepared by spin-coating the precursor solutions on substrates with a spin-coating speed of 1500 rpm at initial 40 s and then 3000 rpm for 20 s. The thicknesses of 9.0, 55, and 144 nm were achieved by spin-coated precursor solutions with concentrations of 0.004, 0.1, and 0.38 M, respectively. Film thicknesses were measured by AFM (Dimensional Icon, Bruker Corporation). If there is no special mention, all of the films used in the characterizations have a thickness of 144 nm. Here, different substrates were used: glass substrates for XRD, the z-cut quartz substrates for OPTPS, UV-Vis-IR spectrophotometer, PL measurements and AFM measurements, ITO glass for current-voltage (I–V) measurement, and SiO$_2$ prism for SFG-VS measurement. The spin-coating process was carried out in the ambient condition with a relative humidity around 20–35%. The OIHP-deposited substrates were then annealed at 100 °C on a hot plate for 30 min. The samples of $[CH_3(CH_2)_{n-1}NH_3]_2PbI_4$ films are denoted as BA$_2$PbI$_4$ (n = 4), HA$_2$PbI$_4$ (n = 6), OA$_2$PbI$_4$ (n = 8), DA$_2$PbI$_4$ (n = 10), DDA$_2$PbI$_4$ (n = 12), and ODA$_2$PbI$_4$ (n = 18), respectively.

**Optical-pump terahertz-probe spectroscopy (OPTPS).** Layout of the OPTPS setup is shown in Supplementary Fig. 14 and Supplementary Note 8. All the OPTPS experiments were conducted by an amplified Ti: sapphire laser system (Coherent Legend) with a repetition rate of 1 kHz, center wavelength of 800 nm, and pulse duration of 45 fs. The laser beam was divided into three portions for the generation and sampling of THz transients, as well as the optical excitation of samples, respectively. The spot diameters on the samples for the THz probe beam and the pump beam were about 1.0 and 4.0 mm, respectively. Free-space electro-optic sampling was used to coherently detect the THz electric-field pulse transmitted through the sample. In order to record the temporal evolution of photo-induced transmission change, the peak of transmitted THz waveform was synchronized with the femtosecond sampling pulse, and simultaneously varied with respect to the optical-pump pulse in time delay. The OPTPS experiments were performed under a dry nitrogen purge at room temperature.

**Temperature-dependent PL spectroscopy.** Steady-state PL emission spectra were measured using an FLS920 fluorescence spectrometer (Edinburgh). A pulsed Xenon flash lamp with an excitation wavelength of 400 nm was used to excite these samples. Besides, a liquid-nitrogen-cooled cryogenic system (Oxford Instruments) with a temperature control module (Mercury iTC) was employed to carry out temperature-dependent measurements.

**SFG-VS experiments.** All the SFG-VS measurements were carried out by a femtosecond broadband sum frequency generation vibrational spectroscopy (BB-SFG-VS) system. Details about instrumental parameters can be found in our recent

publications[69,70]. Briefly, the femtosecond BB-SFG-VS setup was constructed based on a high power regeneration amplifier (Spectra Physics, Spitfire Ace seeded by Mai-Tai SP), which offers 5.0 W output at 800 nm with 13 nm bandwidth, 100 fs pulse duration and 1 kHz repetition rate. A fraction of the output (1.0 W) was passed through a home-built 4-f pulse shaping system to produce a narrowband visible light with a bandwidth of $\leq 5$ cm$^{-1}$ for the SFG Vis probe. A fraction of a pulse (2.0 W) of the output was used for the excitation of a commercial optical parametric amplifier (TOPAS Prime, Spectra Physics) and collinear difference frequency generation system (with AgGaS$_2$ crystal) to generate tunable broadband infrared pulses (2500–15,000 nm) for the SFG probe. The narrowband visible pulse and broadband IR pulse were focused by using plano-convex lens (FL = 750 mm) and off-axis parabolic bare gold mirror (FL = 150 mm) with beam diameters at the sample surface of ~250 and 200 μm, respectively. The incident angles of visible and IR pulses were 60° and 45° relative to the surface normal, respectively. A near-total-internal-reflection geometry was used to collect SFG spectra. To avoid the contribution of the solvent (DMF) to SFG signals of 2D OIHP films, deuterated DMF (DMF-d$_7$) was used for the sample preparation. The resulting spectra were captured by use of a high-sensitivity EMCCD (Andor Newton 970).

**Experiments and data analysis**. The details about other experiments such as current-voltage (I–V) measurements and data analysis can be found in Supplementary Information (Supplementary Note 1, Supplementary Note 2, Supplementary Note 3, Supplementary Note 4, Supplementary Note 5, Supplementary Note 6, Supplementary Note 7, Supplementary Note 8, Supplementary Note 9, Supplementary Fig. 15, Supplementary Table 4).

## Data availability
The data that support the finding in current study are available from the corresponding author upon reasonable request.

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

## Acknowledgements

This work was supported by National Key Research and Development Program of China (2018YFA0208700 and 2017YFA0303500), National Natural Science Foundation of China (21925302, 21633007, 11774354, and 51727806), and Anhui Initiative in Quantum Information Technologies (AHY090000).

## Author contributions

C.L. and J.T. collected the SFG spectra. C.L., J.Y., and F.S. collected the THz spectra. C.L. collected the fluorescence and XRD data. C.L., F.S., and S.Y. processed and analyzed the data. S.Y. and Y.L. designed and supervised the study. C.L., Y.L., and S.Y. wrote the paper. All authors discussed the results of the study.

## Competing interests

The authors declare no competing interests.
