## [Peer Review File · Nature Communications]

Reviewers' comments:

Reviewer #1 (Remarks to the Author):

Manuscript by Le et al presents a spectroscopic study of 2D perovskite material using a powerful combination of advanced spectroscopic methods - THz, SFG, T-dependent PL. Authors address a broad range of materials with different spacer cations and observe clear correlations between the results obtained from different techniques. Authors conclude that the emissive mechanism and carrier mobility are correlated with the disorder in the material, which can be used to adjust the material properties.

Paper clearly carries a high level and novelty due to the methods applied and conclusion draw and potentially can be suitable for Nature Commun.

However, I have strong concerns about the model behind the interpretation of THz data (see below) which I believe should be seriously revised and clarified. Also paper is not particularly carefully written, so will require multiple minor improvements.

Specific remarks:

1. My biggest concern is how authors interpret THz data in terms of mobility. As discussed in SI, THz signal reflects a product of number of charge carriers and their mobility. Therefore, to determine mobility one needs to be absolutely certain about the concentration of photogenerated carriers (and excitons). To estimate this concentration authors briefly introduce in the SI a branching ratio between both and assume it the same for all studied systems. I dont think the assumption that branching ratio is the same for all systems is well justified. I suspect, it actually may not be true, as judging from absorption spectra the localisation of electronic states is different for different materials, so some of them should have more 'excitonic' character while excited states in the others may behave more like free carriers. In this case, the amplitude of THz signal does not represent the mobility but localisation of electronic state. The observed correlations will be still there, but they may point towards somewhat different model. I believe more arguments about the properties of the excited states should be presented (for example based on visible TA experiments) and both interpretations should be discussed.

2) From figure 1 I dont see the evidence that linear regime was reached in THz measurements. No 3 data points lay on a straight line, which probably indicates a presence of Auger recombination. Low-power measurements should be added to confirm linearity. (to be honest intensity dependencies are not very important for the reader and can be moved to SI)

3) the model in figure 1 would look more informative if the timescales are indicated as well as which states are observed by THz spectroscopy. Associated discussion should be clarified.

4) I think statement in line 167 that 'inorganic cages' are 'identical' is misleading. They are not identical in terms of how extended Pb-I structure are and how strong are the couplings between the different structures.

5) When discussing vibrational SFG on perovskite cation - its is a very new topic with only a couple of studies done. I think authors may consider the effects on material interfaces observed in the recent study by Tahara group (Mat. Horizons 2020; DOI: 10.1039/C9MH01394F)

6) Conclusion section is called 'discussion'. There are many similar mistakes throughout the text.

Reviewer #2 (Remarks to the Author):

Review Comments for “Conformational Disorder of Spacer Organic Cations Tunes the Charge Carrier Mobility and Broadband White-Light Emission in Two-Dimensional Organic-Inorganic Perovskites”

Chuanzaho Li, Jin Yang, Fuhai Su, Junjun Tan, Yi Luo, and Shuji Ye

General comments: It is an interesting work; but I'm not sure that it satisfies Nature requirement in terms of broad impacts or scientific understanding; it would be very helpful to correlate mobility values obtained with more conventional electrical measurement. I think the paper is OK and could have narrower applications to characterizing 2-D perovskites, unless the authors can argue otherwise. Here are some detailed points to consider:

- PXRD plots, which can confirm the morphology of the 2-D Perovskites, are nowhere to be seen in the main text.
- The authors should mention the expected peak positions for the grazing-incidence XRD plots shown in the SI. They can use VESTA and models prepared by Kanatzidis, et. al. to verify the peak positions.
- A picture of the devices / films is needed somewhere in the manuscript (ideally in the main body).
- More characterization of charge transport is needed, especially as confirmation to terahertz spectroscopy. For instance, Mott-Gurney or Time of Flight techniques, which are quick and relatively simple, should be used for characterization, which can be found in a number of publications, including a relatively recent one that also deals with some aspect of 2D perovskite (DOI: 10.1021/acsami.9b09381)
- The general claims of this paper are too bold. I don't see this paper as “guiding the design of new broadband white-light emitters”. It could be a good paper for “characterizing gauche defects and alkane chain distortions in 2-D OIHP materials”

Specific Comments

Ln#	Text	Comments
83..	“the value of effective charge carrier mobility can be deduced”	I'm not too familiar with this technique. It would be good if they confirm these mobility values with electronic measurements, such as the Mott-Gurney I-V technique or time-of-flight (ToF) techniques.
132	“thus, the radiative decay from STE potential...”	What about non-radiative recombination, which should be much more dominant unless there are valence “wells” that line up with the momentum (abscissa) wavevectors.
Fig1	Potential Energy vs. Nuclear Coordinate	Fig. 1d seems incorrect to me. The valence band (or “ground state”) should curve downwards. If this is derived from literature, it could be OK. I'd still prefer a more accurate band diagram schematic, though that may be beyond the scope of this work.
134	“it can be naturally anticipated that the mobility difference....self-trapped carriers in 2D OIHP”	With larger lattice spacing between the semiconducting $[\text{PbI}_6]^{4-}$ layers, could the increased potential barriers be expected the primary cause of lower mobility with larger long-chained alkylamines?
147	“To qualitatively analyze this ratio...”	This may be true, in the case of some broadband emission, but it may not be true to claim that it is a “white light” emitter, as seen in Fig. 2. This clearly shows the dominance of the narrowband characteristic fluorescent emission.

Fig2	Fig. 2 (a, c)	The peak positions of the characteristic fluorescence shifts with temperature, which could indicate crystalline phase changes in the material.
198	“...namely, it initially increases ($n \leq 6$) and then increases ($n \geq 6$)...”	It should be “...and then decreases ”.
207	“We therefore conclude....guide the design of new broadband white-light emitters”	The gauche defect energy levels aren't, from my understanding, expected to cause a significant shift in absorption or emission since the energy levels of hydrocarbons are generally outside the band gap of perovskites. Possibly with very long alkylamines this could be true. However, the claim towards white-light emitters is pretty bold, in my opinion, and is not evinced by the PL spectra in this paper. Aromatic and / or thiolated / silanized amines are probably in a better path to broadband emission.

Reviewer #3 (Remarks to the Author):

Comments:

Two-dimensional (2D) organic-inorganic hybrid perovskites (OIHPs) have emerged as more intrinsically stable materials for solar cells. The organic cations spacer provides an important knob to tune the functionalities of the materials. However, how the physical and chemical nature of the organic cations affects the properties of 2D-OIHPs and devices is rarely studied. In this manuscript, Ye and coworkers employed symmetry sensitive spectroscopic tool - SFG to study the structure of a series of 2D-OIHPs with different chain lengths for the cations on CaF₂ substrate. The intensity ratio between CH₃ and CH₂ was used as an indicator to demonstrate the structural ordering, which was discussed with the charge carrier mobility and broad & narrow band light emission. The correlation between the structural ordering and the other physical properties of the 2D-OIHPs seems obvious. However, some major issues need to be addressed before the manuscript to be accepted for publication in Nat. Comm.

- 1), It is well known in SFG community that intensity of a resonance depends both on the long range ordering of the film and also the average orientation of the functional group with respect to the polarization of the incident electric fields. To compare the intensity ratios for samples with different chain lengths, the average orientations of the molecules have to be measured independently. There seems no discussion about this issue in the manuscript;
- 2), related, the macroscopic thicknesses of the films on CaF₂ substrates were not shown. There have been many reports that different modes of SFG spectrum may show different film thickness dependencies. The trend shown in Fig. 3a and 3b may also accidentally be due to the different film thicknesses. This issue should be discussed.
- 3), look closely into Fig. 3a, there seems a clear peak frequency shift between the spectrum for n=12 and n=18. Is this shift real? If yes, what is the reason? Another question regarding the spectral change, it seems the fermi resonances of the CH₃ mode of different samples also change with chain lengths. Is there any explanation for that?
- 4), from the Temperature-dependent fluorescence map of 2D OIHP films (Fig. 2a-c), it seems n=6 film has completely different mechanism than the other films. Is this feature reproducible for different batches of samples?
- 5) A few other minor comments:
 - A), the intensity/scale bar in Fig. 2a-c is missing.
 - B), Page 6, line 111, "two-phonon" should be "two-photon"

Reviewer #4 (Remarks to the Author):

The authors investigated 2D perovskite materials bearing organic cations with different alkyl chain lengths, and found a correlation between the charge carrier mobility/emitting property and the conformational order of the alkyl spacer. The observation is interesting and meaningful for the application of 2D perovskites, while some concerns need to be carefully addressed before I recommend its publication on Nature Communications.

1. First of all, it is important to make clear the charge transport in which direction is of more interest in real devices. For thin film devices like solar cells and LEDs, the charge transport along the film normal (i.e. perpendicular to the film plane) is more important, while for FETs, that along the film plane plays a major role. For 2D perovskites, the in-plane mobility of the charge carriers is expected to be much higher than the interlayer ones. The ideal case for its application in solar cells and LEDs would be that the 2D layers are packed perpendicular to the substrate, which is apparently not the case in the present study. According to their XRD data (SI figure 8), the 2D layers are most likely packed parallel to the substrate surface. OPTPS is a technique to determine the charge carrier mobility along the film plane. That means the authors were probing the in-plane charge carrier mobility of the 2D perovskite sheets, which is of course not so sensitive to the spacer length. Therefore, it is meaningless to compare the present results to those in the references, e.g. Ref 17, which applied another method to extract the charge carrier mobility along the film normal and hence drew a conclusion that spacer length plays a crucial role. I would

recommend the authors to perform similar measurements to complement their physical model.

2. SFG technique generally requires non-centrosymmetry of the studied system. Since the 2D perovskite sheet possesses alkyl chains on both sides, extending oppositely to each other, I wonder if SFG is only sensitive to the molecules on the top surface and on the film/substrate interface. What was the thickness of the films? Was there any optical interference between the top and bottom interfaces? Could the authors prepare monolayer or bilayer films?

3. How about the surface roughness and spatial homogeneity of the film and did they affect the alkyl chain conformation? AFM measurements might be helpful.

4. I notice that they used different substrates for different measurements. Some might be hydrophilic while some others are hydrophobic. Would the substrate surface property affect the organization pattern of the perovskite materials? Experimental evidence is needed.

5. The tilt angle of the alkyl chain could also influence the methyl/methylene signal ratio. Have the authors measured the SFG spectra with a different polarization combination e.g. ppp? One needs to clarify if the tilt angle is similar for the six samples.

6. The authors claimed that the presence of gauche defect and local chain distortion can form a potential well that can underpin the localization and self-trapping of charge carriers. Considering the relatively large lattice constant along the a-axis of hybrid perovskite, the distance between neighboring organic cations could be too far for the alkyl chains to be densely packed with each other. That might be why all the six samples showed a substantial signal of gauche defects, quite different to the situations reported for SAMs and Langmuir films of alkyl ligands. In such a case, I doubt a gauche defect itself could directly serve as a trap of charge carriers. I believe there must be a slight local distortion of the perovskite lattice in correspondence to the formation of gauche defects in the alkyl chain. Theoretical simulations are suggested, if time permits.

Responses to the report of Reviewer 1:

General Comments: *Manuscript by Le et al presents a spectroscopic study of 2D perovskite material using a powerful combination of advanced spectroscopic methods - THz, SFG, T-dependent PL. Authors address a broad range of materials with different spacer cations and observe clear correlations between the results obtained from different techniques. Authors conclude that the emissive mechanism and carrier mobility are correlated with the disorder in the material, which can be used to adjust the material properties. Paper clearly carries a high level and novelty due to the methods applied and conclusion draw and potentially can be suitable for Nature Commun.*

However, I have strong concerns about the model behind the interpretation of THz data (see below) which I believe should be seriously revised and clarified. Also paper is not particularly carefully written, so will require multiple minor improvements.

Author reply: We thank the reviewer very much for his/her appreciation of our work and for stating that “*Paper clearly carries a high level and novelty due to the methods applied and conclusion draw and potentially can be suitable for Nature Commun.*” We are also grateful for his/her insightful comments and suggestions, which have helped us to better understand the experimental findings and to improve the presentation of our results.

Comment 1): *My biggest concern is how authors interpret THz data in terms of mobility. As discussed in SI, THz signal reflects a product of number of charge carriers and their mobility. Therefore, to determine mobility one needs to be absolutely certain about the concentration of photogenerated carriers (and excitons). To estimate this concentration authors briefly introduce in the SI a branching ratio between both and assume it the same for all studied systems. I dont think the assumption that branching ratio is the same for all systems is well justified. I suspect, it actually may not be true, as judging from absorption spectra the localisation of electronic states is different for different materials, so some of them should have more 'excitonic' character while excited states in the others may behave more like free carriers. In this case, the amplitude of THz signal does not represent the mobility but localisation of electronic state. The observed correlations will be still there, but they may point towards somewhat different model. I believe more arguments about the properties of the*

excited states should be presented (for example based on visible TA experiments) and both interpretations should be discussed.

Author reply 1): We thank the reviewer for his/her excellent suggestion. In principle, the photon-to-charge branching ratio ϕ is determined by the exciton binding energy (E_b) of the materials (Gélvez-Rueda et al., *Nat. Commun.* **11**, 1901(2020)). The value of E_b of 2D OIHPs of $[\text{CH}_3(\text{CH}_2)_{n-1}\text{NH}_3]_2\text{PbI}_4$ ($n=4, 6, 8, 9, 10,$ and 12) has been measured by Ishihara et al. in terms of the difference between the bandgap and excitonic absorption peak in the low-temperature (5K) absorption spectrum (Ishihara et al., *Phys. Rev. B* **42**, 11099 (1990)). They found that the value is ~ 320 meV and is independent of the length of the alkyl chains of the organic cations. This conclusion has also been confirmed by theoretical calculation (Quarti et al., *J. Phys. Chem. Lett.* **9**, 3416 (2018)). Therefore, according to previous reports, the photon-to-charge branching ratio can be assumed to be similar for the studied 2D OIHPs in this work.

It is not true that the amplitude of THz signal does not represent the mobility but localization of electronic state. As indicated by many studies (See Supplementary Table 1) and the comments of the fourth reviewer, OPTPS measurement has been demonstrated to be a powerful tool for determining the effective charge carrier mobility ($\phi\mu$) along the film plane. It has been applied to measure the carrier mobility of a series of perovskites. As discussed above, ϕ can be assumed to be similar for all the studied 2D OIHPs in this work. Therefore, the interpretations of THz results is correct.

To address this comment, we have added following contents in main text or SI:

- 1) OPTPS is a technique to determine the charge carrier mobility along the film plane. It is only sensitive to the presence of free charge carriers. Trapped charges or neutral Coulomb-bound electron-hole pairs (excitons) will not be detected³¹. This technique has been demonstrated to be a powerful tool for determining the effective charge carrier mobility along the film plane ($\phi\mu_i$)^{12, 32-37}. Here ϕ is the photon-to-charge branching ratio. It has been applied to measure the carrier mobility of a series of perovskites (See Supplementary Table 1). (Main text, Page 5)
- 2) It is worth mentioning that the difference in the ϕ value may affect the charge-carrier mobility deduced by OPTPS experiments. In principle, the value of ϕ is determined by the

exciton binding energy (E_b) of the materials⁴¹. The value of E_b of 2D OIHPs of $[\text{CH}_3(\text{CH}_2)_{n-1}\text{NH}_3]_2\text{PbI}_4$ ($n=4, 6, 8, 9, 10, \text{ and } 12$) has been measured by Ishihara et al. in terms of the difference between the bandgap and excitonic absorption peak in the low-temperature (5K) absorption spectrum⁴². They found that the value is ~ 320 meV and is independent of the length of the alkyl chains of the organic cations. This conclusion has also been confirmed by theoretical calculation⁴³. According to previous reports, ϕ can be assumed to be similar for all the studied 2D OIHPs in this work. (Main text, Page 7-8)

3) **Supplementary Table 1.** The examples for the charge-carrier mobility values determined by OPTPS measurement. (SI, Page S19)

Composition	Architecture	Mobility ($\text{cm}^2/(\text{V}\cdot\text{s})$)	Reference
MAPbI ₃	Films	35	1
MAPbI ₃	Films	20	2
MAPbI ₃	Single Crystal	600	3
MAPbI _{3-x} ICl _x	Films	33	4
MAPbI _{3-x} ICl _x	Films	27	5
FAPb(Br _x I _{1-x}) ₃ ($0 \leq x \leq 1$)	Films	1-27	6
Cs _{0.17} FA _{0.83} PbI ₃	Films	40	7
Cs _y FA _{1-y} Pb(Br _{0.4} I _{0.6}) ₃	Films	4-21	7, 8
Cs _{0.17} FA _{0.83} Pb(Br _x I _{1-x}) ₃	Films	11-40	7, 9
MASnI ₃	Mesoporous matrix	1.6	10
FASnI ₃	Film	22	11
FASn _{0.5} Pb _{0.5} I ₃	Films	17	12
Cs _{0.25} FA _{0.75} Sn _{0.5} Pb _{0.5} I ₃	Films	14	12
MA _{n-1} PEA ₂ Pb _n I _{3n+1}	Films (2D)	1-25	13
BA _x (FA _{0.83} Cs _{0.17}) _{1-x} Pb(I _{0.6} Br _{0.4}) ₃ ($x \leq 0.8$)	Films	3-15	14
CsPbI ₃	Quantum dots	0.23-0.5	15
PbS	Quantum dots	0.042	15
PbSe	Quantum dots	0.09	15

FAPb(Br _{0.3} I _{0.7}) ₃	Films	2	6
FAPb(Br _{0.4} I _{0.6}) ₃	Films	1	6

Comment 2): From figure 1 I dont see the evidence that linear regime was reached in THz measurements. No 3 data points lay on a straight line, which probably indicates a presence of Auger recombination. Low-power measurements should be added to confirm linearity. (to be honest intensity dependencies are not very important for the reader and can be moved to SI)

Author reply 2): It is a misleading for the fitting curves in the original figure. Actually, when the fluence is less than 45 $\mu\text{J}/\text{cm}^2$, there is a linear correlation between the maximum of $|\Delta T/T|$ and the excitation fluence, as indicated by Supplementary Figure 4. We extract the mobility choosing the data of 25.6 $\mu\text{J}/\text{cm}^2$, which falls in the linear regime. Therefore, this fluence can avoid the unwelcome effects due to the presence of two-photon absorption or nesting effects. Following the reviewer’s suggestion, we replot the Figure 1b and put these data in Supplementary Figure 4. (SI, Page S7)

Supplementary Figure 4. Maximum of $|\Delta T/T|$ as a function of excitation fluences in six 2D OIHP thin films, which saturates at higher fluence ($>45 \mu\text{J}/\text{cm}^2$).

Comment 3): The model in figure 1 would look more informative if the timescales are indicated as well as which states are observed by THz spectroscopy. Associated discussion should be clarified.

Author reply 3): According to the reviewer’s suggestion, we have put the timescales in Figure 1d. Usually, free carrier generation and self-trapping carrier generation show a timescale of hundred femtoseconds (Lindenberg et al., *J. Phys. Chem. Lett.*, 7, 2258 (2016)). In contrast, NE emission state has a lifetime ranging from hundreds of picoseconds to several nanosecond while WE emission state has a lifetime of hundreds of nanoseconds (Deschler et al., *J. Am. Chem. Soc.* 139, 18632 (2017)). OPTPS measurement is only sensitive to the presence of free charge carriers. Trapped charges or neutral Coulomb-bound electron–hole pairs (excitons) will not be detected.

To address this comment, we have added that “OPTPS is a technique to determine the charge carrier mobility along the film plane. It is only sensitive to the presence of free charge carriers. Trapped charges or neutral Coulomb-bound electron–hole pairs (excitons) will not be detected³¹. This technique has been demonstrated to be a powerful tool for determining the effective charge carrier mobility along the film plane ($\phi\mu_i$)^{12, 32-37}. Here ϕ is the photon-to-charge branching ratio. It has been applied to measure the carrier mobility of a series of perovskites (See Supplementary Table 1)” (Main text, Page 5)

Comment 4): *I think statement in line 167 that 'inorganic cages' are 'identical' is misleading. They are not identical in terms of how extended Pb-I structure are and how strong are the couplings between the different structures.*

Author reply 4): We thank the reviewer’s insightful comment. To avoid the misleading, we have revised “It is noticed that these six samples have identical inorganic cage (i.e. the Pb-I layer), the difference in the effective charge carrier mobility and broadband white-light emission should originate from the packing of the organic cations” into “It is noticed that these six samples have identical chemical composition of inorganic cage (i.e. the Pb-I layer), the difference in the effective charge carrier mobility and broadband emission should originate from the packing of the organic cations, which may cause distortion of the inorganic cage”. (Main text, Page 11)

Comment 5): *When discussing vibrational SFG on perovskite cation - its is a very new topic with only a couple of studies done. I think authors may consider the effects on material interfaces observed in the recent study by Tahara group (Mat. Horizons 2020; DOI:*

10.1039/C9MH01394F)

Author reply 5): Thank the reviewer's suggestion. We have added that "SFG has been applied to probe molecular conformation of the organic cations of lead-halide perovskite⁵⁸ and the capping ligands on nanoparticle surfaces^{24, 25, 30}" (Main text, Page 12)

Comment 6): *Conclusion section is called 'discussion'. There are many similar mistakes throughout the text.*

Author reply 6): According to the format of Nature Communication, "Conclusion" section is titled as "Discussion". We have read the manuscript for multiple times and corrected the typos and grammar errors.

Responses to the report of Reviewer 2:

General Comments: *It is an interesting work; but I'm not sure that it satisfies Nature requirement in terms of broad impacts or scientific understanding; it would be very helpful to correlate mobility values obtained with more conventional electrical measurement. I think the paper is OK and could have narrower applications to characterizing 2-D perovskites, unless the authors can argue otherwise.*

Author reply: We are grateful for the reviewer's insightful comments and suggestions, which have helped us to better understand our results and to significantly improve the quality of our manuscript.

Two-dimensional (2D) organic-inorganic hybrid perovskites (OIHPs) are promising candidates for nanophotonics and optoelectronic devices due to its outstanding characteristics. The importance of molecular structures, particularly, structures of spacer organic cations, in determining the properties of 2D OIHPs has been well documented in many studies, which puts huge demands in revealing the structure-property relationship. However, very few experimental techniques can directly probe the molecular structures of the polar organic ligands on the rather rigid inorganic framework within 2D OIHP films. In this study, we applied a powerful combination of advanced spectroscopic methods - THz, SFG, T-dependent PL to investigate the structure-property relationship. We have established a correlation among the conformation of the organic cations, the charge carrier mobility, and broadband emission for the first time. This work provides physical understanding of the important role of organic cation conformation in the properties of 2D-OIHPs and highlights the power of the state-of-the-art SFG-VS technique in revealing structure-property relationship in the perovskite research at the molecular level. This work is certainly of general and broad interest to chemists, materials scientists and engineers, and thus satisfies the requirement of *Nature Communication*. Actually, other reviewers have stated *that* "Paper clearly carries a high level and novelty due to the methods applied and conclusion draw and potentially can be suitable for Nature Commun."

Following the reviewer's suggestions, we have performed the conventional electrical measurement to measure the carrier mobility of 2D OIHPs. It is evident that although out-of-plane charge-carrier mobility determined by Mott-Gurney analysis of the I-V data

curves is about four orders of magnitude smaller than the in-plane mobility given by OPTPS measurements, it follows similar trend of in-plane mobility. Namely, out-of-plane charge-carrier mobility also initially increases ($n \leq 6$) and then decreases ($n \geq 6$) as the chain length of the organic cations increases.

To address this comment, we have added following contents in main text and SI.

- 1) We then determine out-of-plane charge-carrier mobility (μ_o) using Mott–Gurney (M-G) analysis of the I–V data curves. M-G analysis of I–V data (Supplementary Figure 7) has been used to estimate the carrier mobility^{18, 44}. As shown in Fig. 1c and Supplementary Table 3, out-of-plane charge-carrier mobility is about four orders of magnitude smaller than the in-plane mobility and it follows similar trend of in-plane mobility. Namely, out-of-plane charge-carrier mobility also initially increases ($n \leq 6$) and then decreases ($n \geq 6$) as the chain length of the organic cations increases. (Main text, Page 8)
- 2) Fig. 5a and 5b plot the charge carrier mobility against the interlayer distance. It can be seen that both of in-plane and out-of-plane mobilities decrease as the interlayer distance increases for the samples from HA_2PbI_4 ($n=6$) to ODA_2PbI_4 ($n=18$), which is in good agreement with previous studies^{18, 73, 74}. However, the sample of BA_2PbI_4 ($n=4$) does not satisfy such law. In contrast, out-of-plane mobility shows a linear correlation with the ratio of $(\chi_{ss-CH_3}^{(2)}/\chi_{ss-CH_2}^{(2)})/d$ (Fig.5c). It implies that out-of-plane mobility is not only related to the interlayer distance but also dependent on the conformational order of the organic cation. The relationships in Fig. 3c and Fig. 5c reveal that the conformational order of the organic cation plays important roles in regulating the charge carrier mobility and broadband emission in 2D OIHP films. (Main text, Page 16)
- 3) Supplementary Figure 6 (SI, Page S9), Supplementary Figure 7 (SI, Page S10), Supplementary Table 3 (SI, Page S21), Supplementary Note 5 (SI, Page S25-S26).

Comment 1): *PXRD plots, which can confirm the morphology of the 2-D Perovskites, are nowhere to be seen in the main text.*

Author reply 1): According to the reviewer’s suggestion, we have put the XRD patterns of the 2D OIHP thin films in the main text, see Fig.4. (Main text, Page 17)

Fig. 4. a) The XRD results of six 2D OIHP thin films. a) XRD patterns. b) The dependence of interlayer distance (d) on the alkyl chain length of organic cations. The interlayer distance is calculated in terms of the positions of (002) diffraction peak.

Comment 2): *The authors should mention the expected peak positions for the grazing-incidence XRD plots shown in the SI. They can use VESTA and models prepared by Kanatzidis, et. al. to verify the peak positions.*

Author reply 2): According to the reviewer’s suggestion, we have revised “To examine the relationship between mobility and interlayer distance, we determined the interlayer distance of these 2D OIHP thin films by using X-ray diffraction (XRD) techniques” into “To examine the relationship between mobility and interlayer distance, we determined the interlayer distance of these 2D OIHP thin films by using X-ray diffraction (XRD) techniques in terms of the positions of (002) diffraction peak (Fig.4a). The peak positions agree well with the values modeled using the software of VESTA⁷⁰⁻⁷² (Supplementary Figure 15 and Supplementary Table 4). It can be seen that the (002) diffraction peaks display a shift toward smaller angles as the organic length increases (from BA_2PbI_4 ($n=4$) to ODA_2PbI_4 ($n=18$))” (Main text, Page 15)

We have also added Supplementary Figure 15 (The simulated XRD patterns of six 2D OIHPs using VESTA) (SI, Page S18), and Supplementary Table 4 (The comparison of the value of (002) peaks in 2D OIHP films given by experiment and VESTA program) (SI, Page S22).

Comment 3): *A picture of the devices / films is needed somewhere in the manuscript (ideally in the main body).*

Author reply 3): Following the reviewer's suggestion, we have added a picture of the devices/films in the inset of Fig. 1c. (Main text, Page 6). The pictures of the devices with ITO/2D OIHP/Al structure for six samples are given in Supplementary Figure 6. (SI, Page S9)

Comment 4): *More characterization of charge transport is needed, especially as confirmation to terahertz spectroscopy. For instance, Mott-Gurney or Time of Flight techniques, which are quick and relatively simple, should be used for characterization, which can be found in a number of publications, including a relatively recent one that also deals with some aspect of 2D perovskite (DOI: 10.1021/acsami.9b09381)*

Author reply 4): Thank the reviewer's suggestion, we have performed the conventional electrical measurement to measure the carrier mobility of 2D OIHPs. It is evident that although out-of-plane charge-carrier mobility determined by Mott-Gurney analysis of the I-V data curves is about four orders of magnitude smaller than the in-plane mobility given by OPTPS measurements, it follows similar trend of in-plane mobility. Namely, out-of-plane charge-carrier mobility also initially increases ($n \leq 6$) and then decreases ($n \geq 6$) as the chain length of the organic cations increases.

Comment 5): *The general claims of this paper are too bold. I don't see this paper as "guiding the design of new broadband white-light emitters". It could be a good paper for "characterizing gauche defects and alkane chain distortions in 2-D OIHP materials"*

Author reply 5): To avoid the confusion, we have deleted the statements associated with "guiding the design of new broadband white-light emitters".

Specific Comments

Comment 6): *I'm not too familiar with this technique. It would be good if they confirm these mobility values with electronic measurements, such as the Mott-Gurney I-V technique or time-of-flight (ToF) techniques.*

Author reply 6): Following the reviewer’s suggestion, we have determined out-of-plane charge-carrier mobility (μ_o) by Mott–Gurney analysis of the I–V data curves. It is found that although out-of-plane charge-carrier mobility determined by Mott–Gurney analysis of the I–V data curves is about four orders of magnitude smaller than the in-plane mobility given by OPTPS measurements, it follows similar trend of in-plane mobility. Namely, out-of-plane charge-carrier mobility also initially increases ($n \leq 6$) and then decreases ($n \geq 6$) as the chain length of the organic cations increases.

Comment 7): *What about non-radiative recombination, which should be much more dominant unless there are valence “wells” that line up with the momentum (abscissa) wavevectors.*

Author reply 7): Usually, the radiative recombination of STEs may lead to the broadband emission (Kahmann et al., *Nat. Commun.* **11**, 2344(2020)). As indicated by Fig. 2a-c and Supplementary Figure 5, the broadband photoluminescence emission is very weak at room temperature. Therefore, the radiative recombination is small and the non-radiative recombination is dominant at room temperature. However, recent report given by Lindenberg et al. shows that photoluminescence becomes longer and more efficient (~75% PLQE at 20K) at lower temperature, which implies the attenuation of non-radiative decay channels and enhancement of the radiative recombination at lower temperature (Lindenberg et al. *J. Phys. Chem. Lett.* **7**, 2258 (2016)).

To address this comment, we have added “The broadband photoluminescence emission is very weak at room temperature, indicating the radiative recombination is small and the non-radiative recombination is dominant at room temperature. Broadband emission becomes apparent at low temperature, illustrating the attenuation of non-radiative decay channels and enhancement of the radiative recombination³¹”. (Main text, Page 10)

Comment 8): *Fig. 1d seems incorrect to me. The valence band (or “ground state”) should curve downwards. If this is derived from literature, it could be OK. I’d still prefer a more accurate band diagram schematic, though that may be beyond the scope of this work.*

Author reply 8): In physics and geometry, there are two closely related vector spaces:

position space (also real space or coordinate space) and momentum space. The valence band curves upwards in coordinate space while it curves downwards in momentum space. In this work, Fig. 1d is plotted in coordinate space, therefore, the valence band should curve upwards.

Comment 9): *With larger lattice spacing between the semiconducting $[\text{PbI}_6]^{4-}$ layers, could the increased potential barriers be expected the primary cause of lower mobility with larger long-chained alkylamines?*

Author reply 9): It is true that the increased potential barrier is an important cause of lower out-of-plane mobility. For 2D OIHPs, the charge-carrier mobility includes out-of-plane mobility and in-plane mobility. We have discussed the dependence of mobility on the conformation of the organic cations, and the lattice spacing between the semiconducting $[\text{PbI}_6]^{4-}$ layers. It is found that the effective in-plane mobility ($\phi\mu_i$) linearly increases as the trans configuration of alkyl chain (characterized by $\chi_{SS-CH_3}^{(2)}/\chi_{SS-CH_2}^{(2)}$ ratio) increases. However, the linear correlation between in-plane mobility and the reciprocal of the interlayer distance only applies to the samples from HA_2PbI_4 (n=6) to ODA_2PbI_4 (n=18) (Fig. 4b), not to the case of BA_2PbI_4 (n=4). For the out-of-plane mobility, it has a linear dependence of $(\chi_{SS-CH_3}^{(2)}/\chi_{SS-CH_2}^{(2)})/d$, indicating that the out-of-plane mobility not only relies on the interlayer distance, but also the trans configuration of alkyl chain.

To address this comment, we have added that “Fig. 5a and 5b plot the charge carrier mobility against the interlayer distance. It can be seen that both of in-plane and out-of-plane mobilities decrease as the interlayer distance increases for the samples from HA_2PbI_4 (n=6) to ODA_2PbI_4 (n=18), which is in good agreement with previous studies^{18, 73, 74}. However, the sample of BA_2PbI_4 (n=4) does not satisfy such law. In contrast, out-of-plane mobility shows a linear correlation with the ratio of $(\chi_{SS-CH_3}^{(2)}/\chi_{SS-CH_2}^{(2)})/d$ (Fig.5c). It implies that out-of-plane mobility is not only related to the interlayer distance but also dependent on the conformational order of the organic cation. The relationships in Fig. 3c and Fig. 5c reveal that the conformational order of the organic cation plays important roles in regulating the charge carrier mobility and broadband emission in 2D OIHP films” (Main text, Page 16)

Comment 10): *This may be true, in the case of some broadband emission, but it may not be true to claim that it is a “white light” emitter, as seen in Fig. 2. This clearly shows the dominance of the narrowband characteristic fluorescent emission.*

Author reply 10): To avoid the confusion, we have modified “broadband white-light emission” into “broadband emission”.

Comment 11): *The peak positions of the characteristic fluorescence shifts with temperature, which could indicate crystalline phase changes in the material.*

Author reply 11): It is true that the shift of the PL peak positions at low temperature is due to phase transition (Ni et al., *ACS Nano* **11**, 10834 (2017)). Low-temperature PL measurement is used because the broadband emission at room temperature in 2D OIHPs is very weak due to unavoidable presence of scattering. Recently, Wu et al. have confirmed that the existence of STE state can be determined by low-temperature PL spectroscopy even though low-temperature phase transition may shift the PL peak position. They have demonstrated that the result deduced from low-temperature PL emission is consistent with the one measured by transient absorption spectroscopy (TAS) at room temperature (Wu et al., *J. Am. Chem. Soc.* **137**, 2089 (2015)).

To address this comment, we have added that “Low-temperature PL measurement is used because the broadband emission at room temperature in 2D OIHPs is very weak due to unavoidable presence of scattering⁵². Recently, Wu et al. have confirmed that the existence of STE state can be determined by low-temperature PL spectroscopy even though low-temperature phase transition may shift the PL peak position. They have demonstrated that the result deduced from low-temperature PL emission is consistent with the one measured by transient absorption spectroscopy (TAS) at room temperature⁵²”. (Main text, Page 9)

Comment 12): *In line 198 that “...namely, it initially increases ($n \leq 6$) and then increases ($n \geq 6$)...”, It should be “...and then decreases”.*

Author reply 12): We thank the reviewer for his/her carefulness, we have corrected this typos.

Comment 13): *The gauche defect energy levels aren't, from my understanding, expected to cause a significant shift in absorption or emission since the energy levels of hydrocarbons are generally outside the band gap of perovskites. Possibly with very long alkylamines this could be true. However, the claim towards white-light emitters is pretty bold, in my opinion, and is not evinced by the PL spectra in this paper. Aromatic and / or thiolated / silanized amines are probably in a better path to broadband emission.*

Author reply 13): Thank the reviewer's suggestion. To avoid the confusion, we have deleted the statements associated with "guiding the design of new broadband white-light emitters" and modified "broadband white-light emission" into "broadband emission".

Responses to the report of Reviewer 3:

General Comments: *Two-dimensional (2D) organic-inorganic hybrid perovskites (OIHPs) have emerged as more intrinsically stable materials for solar cells. The organic cations spacer provides an important knob to tune the functionalities of the materials. However, how the physical and chemical nature of the organic cations affects the properties of 2D-OIHPs and devices is rarely studied. In this manuscript, Ye and coworkers employed symmetry sensitive spectroscopic tool - SFG to study the structure of a series of 2D-OIHPs with different chain lengths for the cations on CaF₂ substrate. The intensity ratio between CH₃ and CH₂ was used as an indicator to demonstrate the structural ordering, which was discussed with the charge carrier mobility and broad & narrow band light emission. The correlation between the structural ordering and the other physical properties of the 2D-OIHPs seems obvious. However, some major issues needs to be addressed before the manuscript to be accepted for publication in Nat. Comm.*

Author reply: We thank the reviewer for his/her appreciation of our work and for stating that *“Two-dimensional (2D) organic-inorganic hybrid perovskites (OIHPs) have emerged as more intrinsically stable materials for solar cells. ...The correlation between the structural ordering and the other physical properties of the 2D-OIHPs seems obvious”*. The insightful comments and suggestions given by the reviewer have helped us to better understand the experimental findings and to improve the presentation of our results.

Comment 1): *It is well known in SFG community that intensity of a resonance depends both on the long range ordering of the film and also the average orientation of the functional group with respected to the polarization of the incident electric fields. To compare the intensity ratios for samples with different chain lengths, the average orientations of the molecules have to be measured independently. These seems no discussion about this issue in the manuscript;*

Author reply 1): We thank reviewer for his/her excellent suggestions. Theoretically, the ratio of $\chi_{\text{SS-CH}_3}^{(2)}/\chi_{\text{SS-CH}_2}^{(2)}$ not only depends on the fraction of defected chains but also relies on the orientation of the hydrocarbon chain. However, previous studies have demonstrated that a

dramatic reduction in the order ratio without introduction of more defects must require a significant variation of the orientation angle, which is not the case of our observations. In our study, it is found that the average orientation of the terminal methyl groups changes very small for these six 2D OIHPs films because the measured ppp and ssp spectral intensity ratio ($\chi_{\text{ppp}}^{(2)}(\text{CH}_3, \text{ss})/\chi_{\text{ssp}}^{(2)}(\text{CH}_3, \text{ss})$) is almost the same (0.9 ± 0.1) (Supplementary Figure 11b, 12d and 13d). Herein, the ratio of $\chi_{\text{ss-CH}_3}^{(2)}/\chi_{\text{ss-CH}_2}^{(2)}$ dominated by the defected chains and not the chain orientation.

To address the comment, we have added following contents in main text and SI.

- 1) There are some concerns that the $\chi_{\text{ss-CH}_3}^{(2)}/\chi_{\text{ss-CH}_2}^{(2)}$ ratio may be affected by the chain orientation and the film thickness. Theoretically, the ratio of $\chi_{\text{ss-CH}_3}^{(2)}/\chi_{\text{ss-CH}_2}^{(2)}$ not only depends on the fraction of defected chains but also relies on the orientation of the hydrocarbon chain. However, previous studies have demonstrated that a dramatic reduction in the order ratio without introduction of more defects must require a significant variation of the orientation angle (see Supplementary Note 8)^{30, 67-69}, which is not the case of our observations. In our study, it is found that the average orientation of the terminal methyl groups changes very small for these six 2D OIHP films because the measured ppp and ssp spectral intensity ratio ($\chi_{\text{ppp}}^{(2)}(\text{CH}_3, \text{ss})/\chi_{\text{ssp}}^{(2)}(\text{CH}_3, \text{ss})$) is almost the same (0.9 ± 0.1) (Supplementary Figure 11b, 12d and 13d). Herein, the ratio of $\chi_{\text{ss-CH}_3}^{(2)}/\chi_{\text{ss-CH}_2}^{(2)}$ is dominated by the defected chains and not the chain orientation. (Main text, Page 14)
- 2) Supplementary Note 8. The possible influence of chain orientation and film thickness on the $\chi_{\text{ss-CH}_3}^{(2)}/\chi_{\text{ss-CH}_2}^{(2)}$ ratio. (SI, Page S27-S28)

Comment 2): *Related, the macroscopic thicknesses of the films on CaF₂ substrates were not shown. There have been many reports that different modes of SFG spectrum may show different film thickness dependencies. The trend shown in Fig. 3a and 3b may also accidentally be due to the different film thicknesses. This issue should be discussed.*

Author reply 2): To evaluate the influence of the film thickness on the $\chi_{\text{ss-CH}_3}^{(2)}/\chi_{\text{ss-CH}_2}^{(2)}$

ratio, we measure the SFG spectra of the films (on SiO₂ prisms) with the thicknesses of 9.0 nm, 55 nm, and 144 nm. It is found that although the $\chi_{SS-CH_3}^{(2)}/\chi_{SS-CH_2}^{(2)}$ ratio has a little variation for different thicknesses, the dependence of the $\chi_{SS-CH_3}^{(2)}/\chi_{SS-CH_2}^{(2)}$ ratio on the alkyl chain length follows similar trend for the films with the same thickness.

To address this comment, we have added that “To evaluate the influence of the film thickness, we further measure the SFG spectra of the films with the thickness of 9.0 nm and 55 nm (Supplementary Figure 12 and 13). It is found that although the $\chi_{SS-CH_3}^{(2)}/\chi_{SS-CH_2}^{(2)}$ ratio has a little variation for different thicknesses, the dependence of the $\chi_{SS-CH_3}^{(2)}/\chi_{SS-CH_2}^{(2)}$ ratio on the alkyl chain length follows similar trend for the films with the same thickness” (Main text, Page 14)

We have added the SFG results of the films with different thickness in Supplementary Figure 11 (144 nm) (SI, Page S14), Supplementary Figure 12 (9.0 nm) (SI, Page S15), Supplementary Figure 13 (55 nm) (SI, Page S16).

Comment 3): *Look closely into Fig. 3a, there seems a clear peak frequency shift between the spectrum for n=12 and n=18. Is this shift real? If yes, what is the reason? Another question regarding the spectral change, it seems the fermi resonances of the CH₃ mode of different samples also change with chain lengths. Is there any explanation for that?*

Author reply 3): In the previous version of the manuscript, the SFG spectra were collected from the films deposited on CaF₂ prisms using an ICCD (Andorstar 734) and IR peak positions were not calibrated using standard samples. The frequency shift comes from the small shift of the femtosecond 800 nm pulse at different days. Therefore, it is not a real shift. The broad peak in the Fermi resonant region is contributed by CH₂ asymmetric stretch, Fermi resonance of CH₃ group and CH₃ asymmetric stretch and thus change with chain lengths. In the revised manuscript, we collected the SFG spectra from the films deposited on SiO₂ prisms using a high-sensitivity EMCCD (Andor Newton 970) and calibrated the IR peak positions using standard polystyrene film.

Comment 4): *From the Temperature-dependent fluorescence map of 2D OIHP films (Fig. 2a-c), it seems n=6 film has completely different mechanism than the other films. Is this feature reproducible for different batches of samples?*

Author reply 4): The feature of temperature-dependent fluorescence is reproducible for different batches of samples. Such spectral features are consistent with previous reports. For example, the film of n=6 does not have obvious broadband PL peak at above 100K, while the films of n=4 and n=12 have strong broadband PL signal at 100K (Booker et al., *J. Am. Chem. Soc.* **139**, 18632 (2017); Booker et al., *Adv. Optical Mater.* **6**, 1800616 (2018); Wu et al., *J. Am. Chem. Soc.* **137**, 2089 (2015)).

To address this comment, we have added that “Such spectra features are consistent with previous reports, in which the film of n=6 does not have obvious broadband PL peak at above 100K, while the films of n=4 and n=12 have strong broadband PL signal at 100K⁵²⁻⁵⁴” (Main text, Page 9-10)

Comment 5): A few other minor comments: A), the intensity/scale bar in Fig. 2a-c is missing. B), Page 6, line 111, “two-phonon” should be “two-photon”

Author reply 5): We thank the reviewer for his/her carefulness, we have added the scale bar in Fig. 2a-c and corrected this typos.

Responses to the report of Reviewer 4:

General Comments: *The authors investigated 2D perovskite materials bearing organic cations with different alkyl chain lengths, and found a correlation between the charge carrier mobility/emitting property and the conformational order of the alkyl spacer. The observation is interesting and meaningful for the application of 2D perovskites, while some concerns need to be carefully addressed before I recommend its publication on Nature Communications.*

Author reply: We thank the reviewer for such an excellent summary of our paper, and for his/her appreciation of our findings. The insightful comments and suggestions given by the reviewer have helped us to better understand our results and to significantly improve the quality of our manuscript.

Comment 1): *First of all, it is important to make clear the charge transport in which direction is of more interest in real devices. For thin film devices like solar cells and LEDs, the charge transport along the film normal (i.e. perpendicular to the film plane) is more important, while for FETs, that along the film plane plays a major role. For 2D perovskites, the in-plane mobility of the charge carriers is expected to be much higher than the interlayer ones. The ideal case for its application in solar cells and LEDs would be that the 2D layers are packed perpendicular to the substrate, which is apparently not the case in the present study. According to their XRD data (SI figure 8), the 2D layers are most likely packed parallel to the substrate surface. OPTPS is a technique to determine the charge carrier mobility along the film plane. That means the authors were probing the in-plane charge carrier mobility of the 2D perovskite sheets, which is of course not so sensitive to the spacer length. Therefore, it is meaningless to compare the present results to those in the references, e.g. Ref 17, which applied another method to extract the charge carrier mobility along the film normal and hence drew a conclusion that spacer length plays a crucial role. I would recommend the authors to perform similar measurements to complement their physical model.*

Author reply 1): We thank the reviewer for these excellent suggestion. We have performed the conventional electrical measurement to measure the out-of-plane mobility of 2D OIHPs by Mott–Gurney analysis of the I–V data curves. It is found that although out-of-plane

charge-carrier mobility is about four orders of magnitude smaller than the in-plane mobility given by OPTPS measurements, it follows similar trend of in-plane mobility. Namely, out-of-plane charge-carrier mobility also initially increases ($n \leq 6$) and then decreases ($n \geq 6$) as the chain length of the organic cations increases. We plot the charge carrier mobility against the interlayer distance (Fig.5a and 5b). It can be seen that both of in-plane and out-of-plane mobilities decrease as the interlayer distance increases for the samples from HA₂PbI₄ ($n=6$) to ODA₂PbI₄ ($n=18$). However, the sample of BA₂PbI₄ ($n=4$) does not satisfy such law. In contrast, out-of-plane mobility shows a linear correlation with the ratio of $(\chi_{SS-CH_3}^{(2)}/\chi_{SS-CH_2}^{(2)})/d$ (Fig.5c). It implies that out-of-plane mobility is not only related to the interlayer distance but also dependent on the conformational order of the organic cation. The results in Fig. 3c and Fig. 5c reveal that the conformational order of the organic cation plays important role in regulating the charge carrier mobility and broadband emission in 2D OIHP films.

To address this comment, we have added following contents in main text and SI.

- 1) We then determine out-of-plane charge-carrier mobility (μ_o) using Mott–Gurney (M-G) analysis of the I–V data curves. M-G analysis of I–V data (Supplementary Figure 7) has been used to estimate the carrier mobility^{18, 44}. As shown in Fig. 1c and Supplementary Table 3, out-of-plane charge-carrier mobility is about four orders of magnitude smaller than the in-plane mobility and it follows similar trend of in-plane mobility. Namely, out-of-plane charge-carrier mobility also initially increases ($n \leq 6$) and then decreases ($n \geq 6$) as the chain length of the organic cations increases. (Main text, Page 8)
- 2) Fig. 5a and 5b plot the charge carrier mobility against the interlayer distance. It can be seen that both of in-plane and out-of-plane mobilities decrease as the interlayer distance increases for the samples from HA₂PbI₄ ($n=6$) to ODA₂PbI₄ ($n=18$), which is in good agreement with previous studies^{18, 73, 74}. However, the sample of BA₂PbI₄ ($n=4$) does not satisfy such law. In contrast, out-of-plane mobility shows a linear correlation with the ratio of $(\chi_{SS-CH_3}^{(2)}/\chi_{SS-CH_2}^{(2)})/d$ (Fig.5c). It implies that out-of-plane mobility is not only related to the interlayer distance but also dependent on the conformational order of the organic cation. The relationships in Fig. 3c and Fig. 5c reveal that the conformational order of the organic cation plays important roles in regulating the charge carrier mobility and

broadband emission in 2D OIHP films. (Main text, Page 16)

- 3) Supplementary Figure 6 (SI, Page S9), Supplementary Figure 7 (SI, Page S10), Supplementary Table 3 (SI, Page S21), Supplementary Note 5 (SI, Page S25-S26).

Comment 2): *SFG technique generally requires non-centrosymmetry of the studied system. Since the 2D perovskite sheet possesses alkyl chains on both sides, extending oppositely to each other, I wonder if SFG is only sensitive to the molecules on the top surface and on the film/substrate interface. What was the thickness of the films? Was there any optical interference between the top and bottom interfaces? Could the authors prepare monolayer or bilayer films?*

Author reply 2): SFG is a second-order coherent nonlinear optical technique that is quite sensitive to molecular symmetry. SFG signals solely arise from non-centrosymmetric motifs. Because of the ordered arrangement of the lattice of 2D OIHPs, the SFG signals of OIHPs are actually similar to the case of Quartz, in which SFG signals come from not only the molecules at surface and interface, but also the bulk molecules. Such selection rules make SFG show powerful potential in the research of nanomaterials. In fact, SFG has been applied to probe molecular conformation of the organic cations of lead-halide perovskite (Tahara et al. Mater. Horiz. **7**, 1348(2020)) and the capping ligands on nanoparticle surfaces (Weeraman et al., J. Am. Chem. Soc. **128**, 14244 (2006); Frederick et al., J. Am. Chem. Soc. **133**, 7476 (2011); Zhang et al., J. Phys. Chem. Lett. **6**, 2170 (2015)). To evaluate the influence of the film thickness on the SFG spectra, we have measured the SFG spectra of the films with the thicknesses of 9.0 nm, 55 nm and 144 nm. The thickness of 2D OIHP films were determined by AFM measurements. It is found that although the $\chi_{SS-CH_3}^{(2)}/\chi_{SS-CH_2}^{(2)}$ ratio has a little variation for different thicknesses, the dependence of the $\chi_{SS-CH_3}^{(2)}/\chi_{SS-CH_2}^{(2)}$ ratio on the alkyl chain length follows similar trend for the films with the same thickness.

To address this comment, we have added following contents in main text and SI.

- 1) Because of the ordered arrangement of the lattice of 2D OIHPs, the SFG signals of OIHPs are actually similar to the case of Quartz, in which SFG signals come from not only the molecules at surface and interface, but also the bulk molecules. Such selection rules make

SFG show powerful potential in the research of nanomaterials. In fact, SFG has been applied to probe molecular conformation of the organic cations of lead-halide perovskite⁵⁸ and the capping ligands on nanoparticle surfaces^{24, 25, 30}. (Main text, Page 12)

- 2) There are some concerns that the $\chi_{ss-CH_3}^{(2)}/\chi_{ss-CH_2}^{(2)}$ ratio may be affected by the chain orientation and the film thickness. Theoretically, the ratio of $\chi_{ss-CH_3}^{(2)}/\chi_{ss-CH_2}^{(2)}$ not only depends on the fraction of defected chains but also relies on the orientation of the hydrocarbon chain. However, previous studies have demonstrated that a dramatic reduction in the order ratio without introduction of more defects must require a significant variation of the orientation angle (see Supplementary Note 8)^{30, 67-69}, which is not the case of our observations. In our study, it is found that the average orientation of the terminal methyl groups changes very small for these six 2D OIHP films because the measured ppp and ssp spectral intensity ratio ($\chi_{ppp}^{(2)}(CH_3, ss)/\chi_{ssp}^{(2)}(CH_3, ss)$) is almost the same (0.9 ± 0.1) (Supplementary Figure 11b, 12d and 13d). Herein, the ratio of $\chi_{ss-CH_3}^{(2)}/\chi_{ss-CH_2}^{(2)}$ is dominated by the defected chains and not the chain orientation. To evaluate the influence of the film thickness, we further measure the SFG spectra of the films with the thicknesses of 9 nm and 55 nm (Supplementary Figure 12 and 13). It is found that although the $\chi_{ss-CH_3}^{(2)}/\chi_{ss-CH_2}^{(2)}$ ratio has a little variation for different thicknesses, the dependence of the $\chi_{ss-CH_3}^{(2)}/\chi_{ss-CH_2}^{(2)}$ ratio on the alkyl chain length follows similar trend for the films with the same thickness. (Main text, Page 14)
- 3) Supplementary Figure 11 (The SFG results of the films with a thickness of 144 nm) (SI, Page S14), Supplementary Figure 12 (The SFG results of the films with a thickness of 9.0 nm) (SI, Page S15), Supplementary Figure 13 (The SFG results of the films with a thickness of 55 nm) (SI, Page S16), Supplementary Note 8. The possible influence of chain orientation and film thickness on the $\chi_{ss-CH_3}^{(2)}/\chi_{ss-CH_2}^{(2)}$ ratio (SI, Page S27-28).

Comment 3): *How about the surface roughness and spatial homogeneity of the film and did they affect the alkyl chain conformation? AFM measurements might be helpful.*

Author reply 3): We prepared the films with different thickness and measured the surface roughness and spatial homogeneity of the films with a thickness of 144 nm using AFM measurements. As shown below figures, the roughness is 2.0 ± 1.0 nm for the films of $n=4-8$ and 10.0 ± 2.0 nm for the films of $n=10-18$. There is no correlation between the alkyl chain conformation and the roughness (Supplementary Figure 14).

Supplementary Figure 14. The surface roughness and spatial homogeneity of the films determined by AFM measurements. The roughness is 2.0 ± 1.0 nm for the films of $n=4-8$ and 10.0 ± 2.0 nm for the films of $n=10-18$. The correlation between the alkyl chain conformation and the roughness is not observed.

Comment 4): *I notice that they used different substrates for different measurements. Some might be hydrophilic while some others are hydrophobic. Would the substrate surface property affect the organization pattern of the perovskite materials? Experimental evidence is needed.*

Author reply 4): To evaluate the influence of the substrates on the SFG spectra, we prepared differently thick films on SiO₂ prisms. It is found that although the $\chi_{SS-CH_3}^{(2)}/\chi_{SS-CH_2}^{(2)}$ ratio has a little variation for different thicknesses and substrates, the dependence of the $\chi_{SS-CH_3}^{(2)}/\chi_{SS-CH_2}^{(2)}$ ratio on the alkyl chain length follows similar trend for the films with the same thickness. See Supplementary Figure 11 (The SFG results of the films with a thickness of 144 nm) (SI, Page S14), Supplementary Figure 12 (The SFG results of the films with a thickness of 9.0 nm) (SI, Page S15), Supplementary Figure 13 (The SFG results of the films with a thickness of 55 nm) (SI, Page S16).

Comment 5): *The tilt angle of the alky chain could also influence the methyl/methylene signal ratio. Have the authors measured the SFG spectra with a different polarization combination e.g. ppp? One needs to clarify if the tilt angle is similar for the six samples.*

Author reply 5): We thank reviewer for his/her excellent suggestions. Theoretically, the ratio of $\chi_{SS-CH_3}^{(2)}/\chi_{SS-CH_2}^{(2)}$ not only depends on the fraction of defected chains but also relies on the orientation of the hydrocarbon chain. However, previous studies have demonstrated that a dramatic reduction in the order ratio without introduction of more defects must require a significant variation of the orientation angle, which is not the case of our observations. In our study, it is found that the average orientation of the terminal methyl groups changes very small for these six 2D OIHPs films because the measured ppp and ssp spectral intensity ratio ($\chi_{ppp}^{(2)}(CH_3, ss)/\chi_{ssp}^{(2)}(CH_3, ss)$) is almost the same (0.9 ± 0.1) (Supplementary Figure 11b, 12d and 13d). Herein, the ratio of $\chi_{SS-CH_3}^{(2)}/\chi_{SS-CH_2}^{(2)}$ dominated by the defected chains and not the chain orientation.

To address the comment, we have added following contents in main text and SI.

- 1) There are some concerns that the $\chi_{SS-CH_3}^{(2)}/\chi_{SS-CH_2}^{(2)}$ ratio may be affected by the chain orientation and the film thickness. Theoretically, the ratio of $\chi_{SS-CH_3}^{(2)}/\chi_{SS-CH_2}^{(2)}$ not only depends on the fraction of defected chains but also relies on the orientation of the hydrocarbon chain. However, previous studies have demonstrated that a dramatic

reduction in the order ratio without introduction of more defects must require a significant variation of the orientation angle (see Supplementary Note 8)^{30, 67-69}, which is not the case of our observations. In our study, it is found that the average orientation of the terminal methyl groups changes very small for these six 2D OIHP films because the measured ppp and ssp spectral intensity ratio ($\chi_{ppp}^{(2)}(CH_3, ss)/\chi_{ssp}^{(2)}(CH_3, ss)$) is almost the same (0.9 ± 0.1) (Supplementary Figure 11b, 12d and 13d). Herein, the ratio of $\chi_{ss-CH_3}^{(2)}/\chi_{ss-CH_2}^{(2)}$ is dominated by the defected chains and not the chain orientation. (Main text, Page 14)

- 2) Supplementary Note 8. The possible influence of chain orientation and film thickness on the $\chi_{ss-CH_3}^{(2)}/\chi_{ss-CH_2}^{(2)}$ ratio. (SI, Page S27-S28)

Comment 6): *The authors claimed that the presence of gauche defect and local chain distortion can form a potential well that can underpin the localization and self-trapping of charge carriers. Considering the relatively large lattice constant along the a-axis of hybrid perovskite, the distance between neighboring organic cations could be too far for the alkyl chains to be densely packed with each other. That might be why all the six samples showed a substantial signal of gauche defects, quite different to the situations reported for SAMs and Langmuir films of alkyl ligands. In such a case, I doubt a gauche defect itself could directly serve as a trap of charge carriers. I believe there must be a slight local distortion of the perovskite lattice in correspondence to the formation of gauche defects in the alkyl chain. Theoretical simulations are suggested, if time permits.*

Author reply 6): We agree with the reviewer that "there must be a slight local distortion of the perovskite lattice in correspondence to the formation of gauche defects in the alkyl chain". Theoretical simulations are highly desirable to confirm it in the future. Femtosecond visible pump- IR probe will also be helpful to reveal the coupling between inorganic anions and organic cations.

To address this comment, we have added "Of course, the formation of gauche defects in the alkyl chain may cause a slight local distortion of the perovskite lattice. Theoretical simulations are highly desirable to confirm it in the future. Femtosecond visible pump- IR

probe will also be helpful to reveal the coupling between inorganic anions and organic cations”

(Main text, Page 18)

REVIEWERS' COMMENTS

Reviewer #1 (Remarks to the Author):

The revised manuscript address most of my criticism - I recommend paper for publication.

Reviewer #3 (Remarks to the Author):

The authors have addressed most of my comments. However, in the newly added section regarding the intensity ratio of CH₂/CH₃ and the thickness effects, the references are out of date. Many recent works which are directly related to the current study were not acknowledged. For examples, it has been reported that the ordering of molecules with long chain can be affected by many factors, such as substrate-molecule interaction (doi.org/10.1063/1.4921954), phase transition (doi.org/10.1039/C5CP04960A) etc. In current study these factors are also present. Regarding the effect of film thickness, it is rather surprise for the reviewer that the ratio obtained in the current sample system is independent on the film thickness, while others (i.e. doi.org/10.1021/jp202416z, doi.org/10.1063/1.3428668) have shown that different vibrational modes may have different thickness dependent patterns. Some additional discuss is necessary to support their conclusion. When these issues are solved the manuscript should be accepted for publication.

Reviewer #4 (Remarks to the Author):

The authors have addressed my comments and concerns carefully in the revised manuscript. Importantly they have confirmed that the chain conformation can substantially affect both the in-plane and out-plane charge mobility. I am glad to suggest its pulication in Nature Communication.

Responses to the report of Reviewer 1:

Comment: *The revised manuscript address most of my criticism - I recommend paper for publication.*

Author reply: We thank reviewer for the recommendation.

Responses to the report of Reviewer 3:

Comment: *The authors have addressed most of my comments. However, in the newly added section regarding the intensity ratio of CH_2/CH_3 and the thickness effects, the references are out of date. Many recent works which are directly related to the current study were not acknowledged. For examples, it has been reported that the ordering of molecules with long chain can be affected by many factors, such as substrate-molecule interaction (doi.org/10.1063/1.4921954), phase transition (doi.org/10.1039/C5CP04960A) etc. In current study these factors are also present. Regarding the effect of film thickness, it is rather surprise for the reviewer that the ratio obtained in the current sample system is independent on the film thickness, while others (i.e. doi.org/10.1021/jp202416z, doi.org/10.1063/1.3428668) have shown that different vibrational modes may have different thickness dependent patterns. Some additional discuss is necessary to support their conclusion. When these issues are solved the manuscript should be accepted for publication.*

Author reply: Thank the reviewer's suggestion. Following the suggestion of reviewer, we have updated the references. As for the concern on the effect of the film thickness on the SFG intensity, it is true that the vibrational modes of pure organic or polymer films in window geometry may have different thickness dependent patterns. In contrast, the ratio obtained in our sample system shows very small dependence on the film thickness. There are mainly due to following reasons: 1) We employ a near-total-internal-reflection geometry (SiO_2 prisms) to collect the SFG spectra; 2) Symmetric CH_2 and CH_3 modes have almost the same Fresnel coefficient under current experimental geometry; 3) The SFG signals of OIHPs actually come from total asymmetry of the films (including the bulk molecules) because of the ordered arrangement of the lattice of 2D OIHPs. To address this comment, we have added that "It is noted that different vibrational modes of organic or polymer films in window geometry have been reported to be affected by many factors including substrate-molecule interaction⁶⁰, phase

transition⁶¹ and film thickness^{62,63}. However, in our experiments, we employ a near-total-internal-reflection geometry (SiO₂ prisms) to collect SFG spectra at room temperature. Under this experimental geometry, symmetric CH₂ and CH₃ modes have almost the same Fresnel coefficient. In addition, the SFG signals of OIHPs actually come from total asymmetry of the films (including the bulk molecules) because of the ordered arrangement of the lattice of 2D OIHPs. Furthermore, phase transition does not occur to these samples at room temperature. Therefore, the ratio obtained in our sample system shows very small dependence on the film thickness.” (Page 13-14)

Responses to the report of Reviewer 4:

Comment: *The authors have addressed my comments and concerns carefully in the revised manuscript. Importantly they have confirmed that the chain conformation can substantially affect both the in-plane and out-plane charge mobility. I am glad to suggest its publication in Nature Communication.*

Author reply: We thank reviewer for the recommendation.